# Predicting the Performance of Foundation Models via Agreement-on-the-Line

## Abstract

Estimating out-of-distribution performance is critical to safely deploy machine learning models. Recently, Baek et al. showed that the phenomenon "agreement-on-the-line" can be a reliable method for predicting OOD accuracy of models in an ensemble consisting largely of CNNs trained from scratch. However, it is now increasingly common to lightly fine-tune foundation models, and it is unclear whether such fine-tuning is sufficient to produce the needed diversity in models for such agreement-based methods to work properly. In this paper, we develop methods for reliably applying agreement-on-the-line-based performance estimation to fine-tuned foundation models. In particular, we first study the case of fine-tuning a single foundation model, where we extensively study how different types of randomness (linear head initialization, data shuffling, and data subsetting) contribute to the agreement-on-the-line of the resulting model sets. Somewhat surprisingly, we find that it is possible to obtain strong agreement via random initialization of the linear head alone. Next, we find how *multiple* foundation models, pretrained on different data sets but fine-tuned on the same task, also observe agreement-on-the-line. Again rather surprisingly, we demonstrate that these models exhibit some key similarity, that causes them all to lie on the same agreement line. In total, these methods enable reliable and efficient estimation of OOD accuracy for fine-tuned foundation models, without leveraging any labeled OOD data.

## 1 Introduction

Foundation models (FM) approaches, where one first pretrains a large model on open world data then fine-tunes or prompts for a specific downstream task, have proven to be a compelling paradigm for many common machine learning problems. The methods have achieved start-of-the-art results on image classification (Radford et al., 2019; Li et al., 2023; Wang et al., 2023), text classification (Brown et al., 2020), question answering (Devlin et al., 2018), and others, and are particularly noted for their often strong performance on out-of-distribution (OOD) data, that may vary substantially from the data used for fine-tuning (referred to as the in-distribution (ID) data) (Bommasani et al., 2021; Wortsman et al., 2022). Unfortunately, a substantial practical problem arises precisely in this OOD setting: in many cases, one does not have access to labeled OOD data, but only has such data available in *unlabeled* form. Obtaining an explicitly labeled hold-out set for each potential OOD shift is costly and impractical, and thus the field has explored other means for estimating OOD accuracy without labeled data.

Interestingly, across a variety of distribution shift benchmarks, models often observe strong linear correlation between the ID and OOD accuracies of models, a phenomenon dubbed "Accuracy-on-the-line" (ACL) (Miller et al., 2021; Recht et al., 2019; Roelofs et al., 2019). Recently, Baek et al. (2022) empirically demonstrated that for ensembles of deep network classifiers trained from scratch, the rates of ID and OOD agreement also show a strong linear correlation with the same slope and bias. Baek et al. (2022) used this to estimate the accuracies of models in such ensembles, thus providing a simple method for estimating OOD accuracy via unlabeled data alone.

In particular, whenever the ID versus OOD accuracy is strongly linearly correlated, one may estimate the linear trend using agreement without ground truth labels. Unfortunately, the AGL approach requires a *diverse collection* of classifiers over which to compute agreement: classifiers must vary in their predictions. Baek et al. (2022) achieve this diversity through training various models of differ-

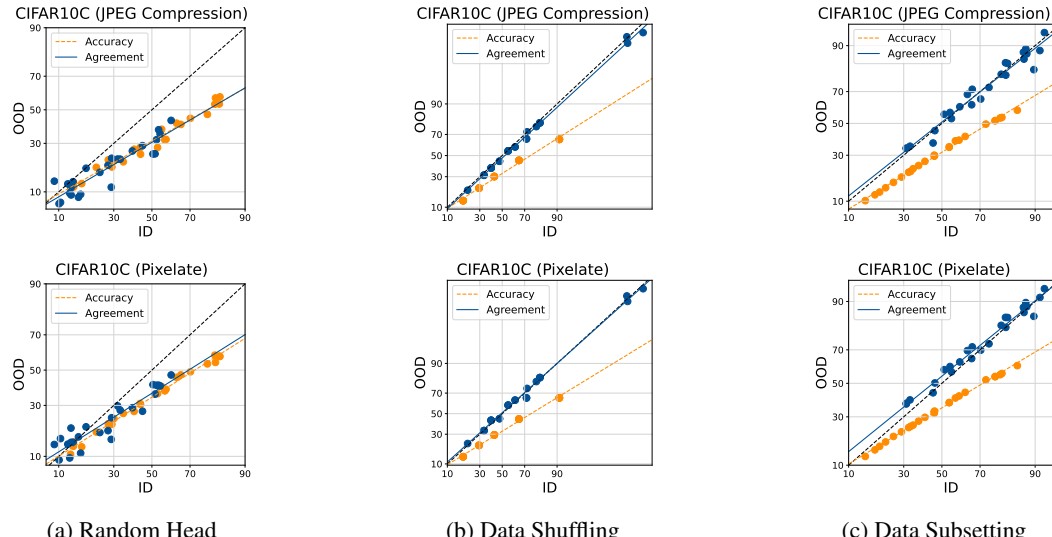

Figure 1: The ID vs OOD trends for accuracy and agreement on the CIFAR10C "JPEG Compression" (top row) and "Pixelate" (bottom row) shifts for linear-probe CLIP models, fine-tuned on CIFAR10 using different sources of randomness, namely - random linear heads, data shuffling, and independent data subsets. Clearly, the use of random linear heads is the only method reliably producing AGL behaviour (i.e. matching the bias and the slope of the two lines)

ent architectures from scratch. However, in the case of fine-tuned FMs, this diversity is seemingly lacking: we often want to *lightly* fine-tune just a single base foundation model for a downstream task, which even after multiple runs would seemingly lead to highly correlated downstream models, thus yielding model sets unsuitable for AGL-based OOD performance estimation.

In this work, we develop methods for extending AGL performance estimation to the setting of FMs, thus enabling practitioners to estimate the OOD performance of fine-tuned models without any labeled data. We first investigate the ability to estimate OOD performance using a *single* base FM. Key to our approach is a detailed empirical study of different types of randomness that we can inject into the fine-tuning process, so as to encourage the needed degree of diversity amongst models. Specifically, we analyze three potential sources of diversity: 1) random linear head initialization; 2) random orderings of the fine-tuning data; and 3) random i.i.d subsets of the fine-tuning data. We find, somewhat surprisingly, that using random linear heads is able to much more reliably induce AGL behavior for the resulting classifiers, despite all settings still resulting in the ACL phenomenon alone. We find that these results hold across multiple different FMs and modalities, holding for CLIP-based image classification, and LLM-based question answering (QA)/text classification tasks. The end result is a simple and straightforward method for evaluating the OOD performance of a fine-tuned FM, applicable to settings where we only want to fine-tune a single such base model.

Second, we analyze the ability of AGL-based method to predict OOD performance when using *multiple* different pretrained FMs. Here the likely problem seems to be opposite to what occurred previously: whereas before we expected to have too little diversity in models, here we encounter a setting where the different base models are pretrained on potentially entirely different data sets, using different architectures and training regiments. We show, however, that this degree of diversity is *also* sufficient for producing AGL behavior. Thus, for settings where multiple FMs exist, they can be all fine-tuned for a given downstream task, and AGL can allow us to estimate their accuracies.

In total, our contributions are as follows:

1. We propose a new state-of-the-art method for unsupervised accuracy estimation under distribution shift when using large pre-trained foundation models that are lightly fine-tuned for specific tasks. Prior works have primarily dealt with models trained from scratch, and hence not directly applicable in this setting. Thus our study is new, computationally tractable, and extremely relevant in today's context.

2. Furthermore, our work leveraging Agreement-on-the-line (AGL) for OOD estimation builds on top of prior work Baek et al. (2022); but extends it in important ways that apply to this new and important setting. The key to making AGL work is obtaining the right ensemble. In Baek et al. (2022), this was done by independently training multiple models from scratch , an unfeasible step for FMs. Our work shows how to side-step this, by systematically identifying a practical method for the same. Specifically, we show that creating an ensemble with randomly initialized linear heads and then fine-tuning, can allow for AGL behavior, and thus ALine methods for unsupervised accuracy estimation, while other similar forms of ensembling (such as data ordering or data subsetting) do not.

3. Besides being of practical relevance, this work also points to several interesting phenomena underlying AGL that go beyond previous knowledge. Prior work Baek et al. (2022) claimed that AGL does not hold for linear models. However, we find the contrary when using pre-trained features. Furthermore, other prior work Miller et al. (2021) suggests that the effective robustness (i.e. the linear fit between ID and OOD accuracy) would change depending on the pretraining data. We find that this is not the case for question answering with different pretrained LLMs. Thus we hope our findings can also advance our understanding of the robustness of ML models, particularly those that leverage foundation models.

This work allows us to substantially expand the set of problems and models for which AGL-based OOD performance estimation is practical, and allows us to leverage much more powerful models for these settings where training models from scratch on tasks of interest is not feasible.

## 2 BACKGROUND AND RELATED WORK

### 2.1 SETUP

Numerous tasks of interest boil down to mapping an input $x \in \mathbb{X}$ to a discrete output $y \in \mathbb{Y}$. In particular, consider a base FM $B : \mathbb{X} \mapsto \mathbb{R}^d$ that we fine-tune to get $f(B) : \mathbb{X} \mapsto \mathbb{Y}$. In this work, we consider a variety of foundation models: GPT2 (Radford et al., 2019), GPT-Neo, OPT (Zhang et al., 2022), Llama2 (Touvron et al., 2023), and CLIP (Radford et al., 2021).

**Fine-tuning.** We have access to labeled data from some distribution $\mathcal{D}_{\text{ID}}$ that we use for obtaining $f(B)$ from $B$. In this work, we consider the following standard fine-tuning procedures.

1. **Linear probing (LP):** Given features $B_\theta$ from the base model $B$, we train a linear head $v$ such that the final classifier maps the score $v^\top B_\phi(x)$ to a predicted class. We randomly initialize $v$ and update $v$ via gradient steps on a suitable loss function such as cross-entropy for classification. We keep the base model parameters frozen, and only update the linear head. We refer to $v$ as either a linear probe (classification), or span prediction head (question answering) depending on the task of interest.

2. **Full fine-tuning (FFT):** Here also we randomly initialize a linear head $v$ and optimize a suitable loss function, but we *update all parameters* of the backbone, i.e. the feature extract $B_\phi$ is also updated. When infeasible to update all parameters natively, we perform *low-rank adaptation* (LoRA) (Hu et al., 2021) which uses trainable rank decomposition matrices to reduce the number of trainable parameters while still effectively updating the feature extractor $B_\phi$. In this work, we do not distinguish between LoRA and FFT as they conceptually achieve the same effect, and seem to show similar empirical trends in our studies.

Several variants of fine-tuning have been proposed, particularly focused on computational efficiency. However, full fine-tuning and linear probing remain the most commonly used approaches.

**OOD performance estimation.** Given access to a labeled validation set from $\mathcal{D}_{\text{ID}}$ and *unlabeled* samples from a potentially different distribution $\mathcal{D}_{\text{ood}}$, our goal is to estimate performance on $\mathcal{D}_{\text{ood}}$. We consider the standard performance metrics for various tasks: Accuracy $\ell_{0\text{-}1} : \mathbb{Y} \mapsto \mathbb{Y}$ for classification and Macro-averaged F1 score $\ell_{\text{F1}} : \mathbb{Y} \mapsto \mathbb{Y}$ for Question Answering. We use $\ell$ to denote the appropriate metric in the context.

## 2.2 BACKGROUND ON OOD ACCURACY ESTIMATION

There is a rich literature on OOD performance estimation, with a variety of proposed approaches. One family of approaches attempts to quantify the degree of distribution shift through data and/or model dependent metrics e.g. uniform convergence bounds using metrics such as $\mathcal{H}$-divergence (Ben-David et al., 2006; Mansour et al., 2009; Cortes et al., 2010; Kuzborskij & Orabona, 2013). However, these approaches only provide upper bounds on the OOD error, and these bounds tend to be loose when evaluated on deep networks used in practice (Miller et al., 2021).

Another line of work looks at leveraging the model's own softmax predictions i.e. the model's confidence to predict the OOD performance (Hendrycks & Gimpel, 2017a; Hendrycks & Dietterich, 2019; Garg et al., 2022; Elsahar & Gallé, 2019; Guillory et al., 2021). Since models are typically overconfident, it is common practice to first calibrate these models using ID validation data to further improve the reliability of such approaches. While these approaches show empirical promise in some settings, they are not expected to work in general and often fail in the presence of large shifts (Garg et al., 2022). There are other heuristic OOD estimation strategies that are reported to work in some datasets such as using performance on auxiliary self-supervised tasks (Schelter et al., 2020; Deng & Zheng, 2021; Deng et al., 2021; Yu et al., 2022) or leveraging characteristics of self-trained models on the OOD data (Yu et al., 2022; Chen et al., 2021).

There has also been growing interest in evaluating the reliability of foundation models in particular, and several distribution shift benchmarks have been proposed to specifically understand the failure modes of large models (Malinin et al., 2022; Tran et al., 2022). However, there has been a lack of study in terms of how well unsupervised performance estimators transfer to large models.

## 2.3 ACCURACY AND AGREEMENT ON THE LINE

In recent work, Baek et al. (2022) propose a different approach for estimating OOD performance, that is empirically reliable across a variety of shifts and outperforms prior approaches. This approach is based on an earlier intriguing observation from (Miller et al., 2021; Recht et al., 2018; 2019; Roelofs et al., 2019; Yadav & Bottou, 2019; Taori et al., 2020; Miller et al., 2020)—there is a strong linear correlation between the ID and OOD performance of models for several distribution shifts. We call this phenomenon "accuracy-on-the-line" (ACL). ACL has been observed for image classification shifts such as some common corruptions on CIFAR10, ImageNetV2, FMoW-WILDS, and question answering shifts such as SQuAD-Shifts. However, ACL does not always hold e.g. Camelyon-WILDS (Miller et al., 2021) and SearchQA (Awadalla et al., 2022) do not show ACL.

While ACL is a striking phenomenon, it does not immediately provide a practical method to estimate OOD performance—computing the slope and bias of the linear correlation requires access to labeled samples from $\mathcal{D}_{\text{ood}}$. Baek et al. (2022) propose to use the *agreement between models* rather than accuracy. Formally, given a pair of models $f_1$ and $f_2$ that map inputs to labels, accuracy and agreement can be defined as

$$\mathsf{Acc}(f_1) = \mathbb{E}_{x,y\sim\mathcal{D}}[\ell(f_1(x), y)], \ \ \mathsf{Agr}(f_1, f_2) = \mathbb{E}_{x,y\sim\mathcal{D}}[\ell(f_1(x), f_2(x))], \quad (1)$$

where $\ell$ is the appropriate performance metric of interest. Note that while accuracy requires access to the ground truth labels $y$, agreement only requires access to unlabeled data and a pair of models. Baek et al. (2022) observed that when ACL is observed i.e. the probit-scaled ID versus OOD accuracies of these models are strongly linearly correlated, then the ID versus OOD probit-scaled agreement of pairs of these models also observe a strong linear correlation with the *same* linear slope and bias. Furthermore, when accuracies do not show a linear correlation, agreements also do not. This phenomenon was called "agreement-on-the-line" (AGL).

Previously, the connection between agreement and accuracy have been explored in-distribution (Jiang et al., 2022; Madani et al., 2004; Nakkiran & Bansal, 2020) and the variance in Bayesian neural networks is often utilized for uncertainty estimation (Gal & Ghahramani, 2016; Lakshminarayanan et al., 2017). Lee et al. (2023) provides some theoretical underpinning of AGL in the regression setting.

Since computing agreement does not require ground truth labels, one can compute the respective slope and bias using OOD unlabeled data, and then estimate the OOD performance from the ID performance measured on ID validation data. We refer the reader to (Baek et al., 2022) for formal

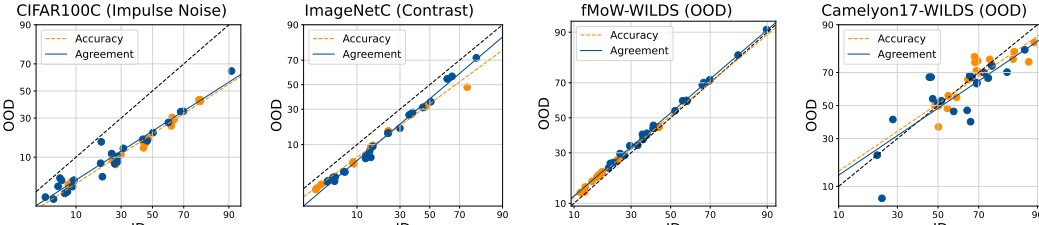

Figure 2: Some examples of datasets where ACL and AGL hold (CIFAR100C, ImageNetC and fMoW-WILDS). Similar to (Baek et al., 2022), we find that ACL doesn't hold with a high correlation for the Camelyon17-WILDS dataset, and consequently neither does AGL.

ALine algorithms (ALine-S and ALine-D) to use AGL for OOD performance estimation (Appendix 9.2). Note that ACL is a pre-requisite for good OOD performance estimation via ALine. However, we can easily detect whether or not ACL holds by simply checking the linear correlations afforded by agreements, and only rely on ALine when agreements show strong linear correlation.

### 2.4 ACL AND AGL: TRAINING FROM SCRATCH VS FINE-TUNING

In this work, we are interested in estimating OOD performance when lightly fine-tuning foundation models. A crucial component for AGL is the *diversity* of the ensemble over which agreements are evaluated. Prior work on AGL has exclusively focused on training from scratch for several epochs, a very different regime from light fine-tuning. If the models are not diverse enough, AGL is bound to fail. As an extreme, consider an ensemble of effectively identical models. Their ID and OOD agreement will always be 1, and there is no linear fit to estimate. In this scenario, it was observed that simply varying the architectures was able to induce the needed diversity for AGL to hold. In contrast, in this work, we focus on how to introduce sufficient diversity during *just* the fine-tuning process which can start from the *same* base foundation model and usually involves far fewer gradient steps than training from scratch.

## 3 PREDICTING OOD PERFORMANCE: SINGLE BASE FOUNDATION MODEL

Our first setting of interest concerns the case where we have a *single* FM that we would like to fine-tune for a given downstream task. Since AGL-methods cannot be applied to a single classifier (requiring a collection of classifiers over which to compute agreement between pairs), we need some method to introduce variability amongst multiple variants of this base model. Such variability can be introduced in many ways, but an overriding concern is that even with some randomness in the fine-tuning process, it may not be enough to overcome the underlying similarities in predictions due to the same base FM.

To address this problem, in this section we evaluate multiple possible sources of diversity in the fine-tuning process, to see what approach (if any) can lead to AGL behavior. Specifically, we analyze three possible methods for introducing diversity into the fine-tuning process (which then lets us create a differentiated collection of classifiers by repeating the fine-tuning process multiple times):

1. **Random linear heads.** Before fine-tuning, we initialize the last layer of the network (i.e., the linear head) randomly, instead of via some zero-shot or pre-specified manner.

2. **Data shuffling.** We present the same data to each model, but shuffle the order for the data differently within each fine-tuning optimization run.

3. **Data subsetting.** We present each model to be fine-tuned with an independent subset of the (ID) fine-tuning data.

For the case of training models from scratch, it is well established that independent data subsetting tends to lead to the greatest diversity of classifiers (Nakkiran & Bansal, 2020). Nonetheless, in this setting we find rather surprisingly, that *just randomizing the linear head* achieves the highest degree

| Source of Diversity | CIFAR10C MAPE (%) | CIFAR10C MAE (%) |
|---|---|---|
| Random linear heads | **15.88** | **5.74** |
| Data shuffling | 74.16 | 22.61 |
| Data subsetting | 25.94 | 7.39 |

Table 1: ALine-D MAE and MAPE for CLIP linear probing on CIFAR10 image classification. Note that the reported MAE and MAPE is averaged across all 19 CIFAR10C evaluated shifts.

of agreement. We show that this finding persists over multiple models, multiple tasks, and indeed multiple modalities entirely.

## 3.1 INVESTIGATIONS ON VLM-BASED IMAGE CLASSIFICATION

**CLIP Linear Probing**   We use CLIP (Radford et al., 2021), specifically the ViT-B/32 model trained on LAION-2B (Schuhmann et al., 2022) for our image classification tasks. Given its well-established 0-shot capabilities, a popular method of fine-tuning CLIP for downstream tasks is to simply employ linear probing on top of the CLIP representation. Thus, in this section we evaluate the OOD performance of similarly obtained ensembles.

**Datasets**   We fine-tune and test our models for several different image classification datasets. We fine-tune models on CIFAR10 (Krizhevsky et al., 2009), and then test on CIFAR10C (Hendrycks & Dietterich, 2019), which contains 50k images with 19 different corruptions, some natural (Snow and Fog), and some synthetic (JPEG compression). We also test on the CIFAR10.1 dataset (Recht et al., 2018), which contains newer images of the same labels. We repeat the same for CIFAR100 (Krizhevsky et al.), ImageNet-1k (Russakovsky et al., 2014) and their respective shifted datasets CIFAR100C, ImageNetC (Hendrycks & Dietterich, 2019), and ImageNetV2 (Recht et al., 2019). We further validate our finding by testing on three real world shifts from the WILDS (FMoW, iWildCam, Camelyon17) (Koh et al., 2021) and the Office-Home (Venkateswara et al., 2017) benchmarks.

**Results**   In Figure 1, we observe the ID and OOD agreements and accuracies of linear probes trained on top of CIFAR10 CLIP representations. One may suspect that in this setting, simply fine-tuning linear models on top of the CLIP representations would agree highly and AGL may break. For example, Baek et al. (2022) has shown previously that AGL is a phenomena that is specific to neural networks e.g. linear models trained on top of the flattened CIFAR10 images do not observe AGL. Indeed, while ACL holds with strong correlation for each of the model collections constructed with the three sources of diversity, AGL does not hold for all model collections. However, AGL interestingly does hold strongly for the case of random head initialization. Thus, contrary to prior findings, even in linear models, when on top of neural network features (in this case CLIP) with the *right type of diversity*, one may observe AGL and use it to predict OOD estimation.

On the contrary, for the other sources of diversity, we observe a consistent trend where agreement is also strongly linearly correlated but generally observe much higher agreement rate OOD. In fact, for all ensembles achieved through data subsetting and data shuffling, the agreement line strictly lies above the accuracy line, i.e. AGL was not observed. In some sense, this is particularly very surprising for linear models. Intuition may suggest that independent data subsetting leads to the greatest diversity as the other sources of diversity optimize over the same convex landscape. Yet even when we distribute the number of epochs trained to achieve a wide spread of ID accuracy models, AGL only holds for models that start at random initializations. The averaged Mean Absolute Percentage Error (MAPE) between the AGL-interpolated and actual OOD accuracies for the CIFAR10C shifts can be found in Table 1, further quantifying these visually apparent results. Figure 2 shows ACL and AGL holding for other datasets over CLIP fine-tuned ensembles obtained with the same random head initialization approach. We refer the reader to Appendix 9.8 for a more exhaustive evaluation.

## 3.2 INVESTIGATIONS ON LLM-BASED TEXT TASKS

We conduct a similar systematic investigation of obtaining a set of fully fine-tuned single base LLMs that are amenable to estimating the OOD performance of any models within that ensemble. Sim-

ilar to CLIP linear probing, we find that AGL cannot consistently be observed without randomly initializing the linear head of the model before performing fine-tuning.

**Models**   We evaluate a collection of 50 fine-tuned models for our experiments in this section. Each model is obtained by fine-tuning from the same checkpoint of a GPT2-Medium. We individually present findings on both these families of models in the following sections. Huggingface links to the base models we trained are in Appendix 9.11.

**Full Fine-tuning**   We fully fine-tune each of our models by attaching a span-prediction head to the pretrained models and fine-tuning the entire network on the ID dataset (SQuAD v1.1). The span-prediction head consists of two linear vectors that estimate the probability with which a token is the start and end of the answer span within the context. Each model is fine-tuned for up to 3 epochs to obtain a sufficient spread of model accuracy. Specifics on hyperparameters for fine-tuning are in Appendix 9.1.

**Datasets**   We study the aforementioned sources of diversity for LLMs for the task of Question Answering (QA). We also include a similar study on the task of Text Classification in Appendix 9.6. Each LLM is fine-tuned on the SQuAD v1.1 dataset (Rajpurkar et al., 2016) for the task of extractive QA. Extractive QA entails finding a single span of text from within the input context that answers the posed question. We evaluate the fine-tuned LLMs on four distribution shifts present in the SQuAD-Shifts (New Wiki, New York Times, Amazon, and Reddit) dataset (Miller et al., 2020). SQuAD-Shifts is a distribution shift in the dataset source. SQuAD builds reading comprehension questions from Wikipedia text, and SQuAD-Shifts replicates this pipeline by using paragraphs obtained from other sources.

**Results**   We find that not all sources of diversity are equally likely to yield a diverse enough ensemble of fine-tuned LLMs. Diversity arising from data shuffling and training on independent data subsets may not always be sufficient enough to yield an ensemble that is amenable to accurately estimating OOD accuracy (see Table 2). Specifically, these sources tend to yield model collections with correlated errors which results in the agreement line often lying above the accuracy line, although the trend is less stark than the one observed with CLIP linear probing for image classification. We refer the reader to Appendix 9.4 to observe these trends on all four shifts within the SQuAD-Shifts dataset. As seen with CLIP linear probing, varying the random initialization of the span head consistently provides sufficient stochasticity during fine-tuning to obtain a suitably diverse ensemble that demonstrates AGL and enables accurate prediction of OOD accuracy.

### 3.3   SUMMARY AND IMPLICATIONS

As a substantial amount of diversity is needed for AGL to hold, it was anticipated that we would not be able to observe AGL for models lightly fine-tuned starting from a single base FM. However, seeing that AGL indeed holds when randomly initializing the linear head during fine-tuning, we show that it is possible to utilize ALine as a metric to compute OOD accuracy. Furthermore, the LLMs were only fully fine-tuned for up to 3 epochs which makes it all the more interesting that fine-tuning would overcome pretraining to achieve diversity. Thus, to a practitioner, when only a single base model is available, the OOD accuracy of another model with the same base can be estimated by randomly initializing a set of models, and fine-tuning. We also refer the reader to Appendix 9.3 for a likewise study of the three sources of diversity when fully fine-tuning CLIP.

| Source of Diversity | SQuAD-Shifts Amazon | | SQuAD-Shifts Reddit | |
|---|---|---|---|---|
| | MAPE (%) | MAE (%) | MAPE (%) | MAE (%) |
| Random Linear Heads | **6.34** | **0.69** | **3.48** | **0.79** |
| Data Shuffling | 10.30 | 4.18 | 9.59 | 4.32 |
| Data Subsetting | 16.21 | 5.2 | 13.94 | 4.71 |

Table 2: ALine-D MAPE(%) and MAE (%) on the SQuAD-Shifts Amazon and Reddit datasets when applied to sets of fully-finetuned models, trained using different sources of randomness

| OOD Dataset | ALine-D | ALine-S | Naive Agr | ATC | AC | DF |
|---|---|---|---|---|---|---|
| SQuAD-Shifts Reddit | **1.20** | 2.60 | 20.21 | 12.74 | 49.25 | 6.09 |
| SQuAD-Shifts Amazon | **1.64** | 3.10 | 20.40 | 15.35 | 51.06 | 7.39 |
| SQuAD-Shifts Nyt | **0.82** | 1.33 | 18.46 | 3.11 | 38.61 | 3.18 |
| SQuAD-Shifts New Wiki | 3.08 | 3.18 | 18.87 | 5.46 | 41.26 | **1.50** |
| Average | **1.68** | 2.55 | 19.48 | 9.16 | 45.04 | 4.54 |
| CIFAR10C (averaged across shifts) | 6.99 | **6.92** | 44.33 | 31.28 | 48.66 | 32.79 |
| CIFAR10.1 (averaged across v4, v6) | **2.42** | 3.03 | 41.52 | 6.48 | 54.57 | 8.51 |
| CIFAR100C (averaged across shifts) | **11.94** | 12.67 | 46.13 | 18.69 | 80.81 | 37.36 |
| ImageNetC (averaged across shifts) | **10.91** | 11.04 | 56.76 | 27.25 | 79.00 | 37.86 |
| ImageNet V2 (averaged across 3 format) | **4.96** | 5.03 | 47.65 | 8.96 | 77.34 | 7.86 |
| fMoW-WILDS (val OOD split) | **2.59** | 2.74 | 83.94 | 9.03 | 44.59 | 5.86 |
| iWildCam-WILDS (val OOD split) | **22.05** | 25.29 | 46.42 | 37.25 | 57.31 | 69.58 |
| Camelyon17-WILDS (val OOD split)* | 9.93 | 10.71 | 19.99 | 18.92 | 24.64 | **7.18** |
| OfficeHome-Art | **9.55** | 13.70 | 45.77 | 29.54 | 76.89 | 27.49 |
| OfficeHome-ClipArt* | 14.60 | 16.23 | 50.81 | 18.22 | 79.44 | **14.29** |
| OfficeHome-Product | **11.10** | 13.98 | 57.28 | 63.35 | 77.13 | 79.97 |
| OfficeHome-Real | **4.80** | 7.20 | 45.16 | 16.43 | 86.44 | 21.93 |

Table 3: The MAPE (%) of predicting OOD performance using ALine and other baseline methods. Evaluations on QA tasks (SQuAD-Shifts) are performed over a set of models finetuned from multiple base FMs (LlaMa, GPT, OPT). Evaluations on the image classification datasets are conducted with CLIP models fine-tuned with linear probing.

## 4 PREDICTING OOD PERFORMANCE: MULTIPLE FOUNDATION MODELS

Alternatively, when multiple base foundation models are accessible, several additional questions arise. Instead of training multiple foundation models with random initialization, if multiple base models heavily pretrained on different data corpora lie on the same ID versus OOD accuracy trend, it's conceivable that these models may also observe the same ID versus OOD agreement trend. However, AGL may potentially fail due to pairs of FMs fine-tuned from different base models disagreeing highly OOD, or models pre-trained on similar corpora observing relatively higher OOD agreement; thus breaking the linear correlation of agreement entirely. All the more, it is unclear whether models heavily pretrained on different text corpora lie on different or similar accuracy lines to begin with. We observe that for certain extractive question-answering shifts, foundation models fine-tuned from a wide range of base models *observe both ACL and AGL.*

**Models** We train 41 models on the extractive QA benchmark SQuAD as in the previous section, and observe their OOD performance to SQuAD-Shifts. We fine-tune OPT-125M, OPT-350M, OPT-1.3B, GPT2-XL, GPT2-Large, GPT2-Medium, GPT2, GPT-Neo-135M, Llama2-7B, Alpaca-7B, and Vicuna-7B to extractive QA. In Appendix 9.7 we perform a similar study on text classification shifts. OPT was pretrained on a wide variety of data including BookCorpus (Zhu et al., 2015), Stories (Trinh & Le, 2018), a subset of PILE (Gao et al., 2020), CCNews v2 corpus, and PushShift.io Reddit (Baumgartner et al., 2020). Similarly, GPT2 was pretrained on BookCorpus while GPT-Neo was trained on PILE. Llama2 was trained on an undisclosed set of publicly available data. Sprouting from Llama2, Alpaca is additionally trained from Llama2 on instruction-following demonstrations while Vicuna is additionally trained from Llama2 on user-shared conversations from ShareGPT.

### 4.1 RESULTS

We investigate the behavior of an ensemble of foundation models fine-tuned from diverse base models on SQuAD to all SQuAD-Shifts datasets in Figure 3. We first make the observation that base LLM models pretrained on different sources of text corpus lead to fine-tuned models that lie on the *same linear trend in accuracy* on SQuAD. This is in contrast to how previous works benchmarking the performance of foundation models on image classification tasks (Radford et al., 2021; Taori et al., 2020) have indicated that models heavily pretrained on differerent image corpus may lie on

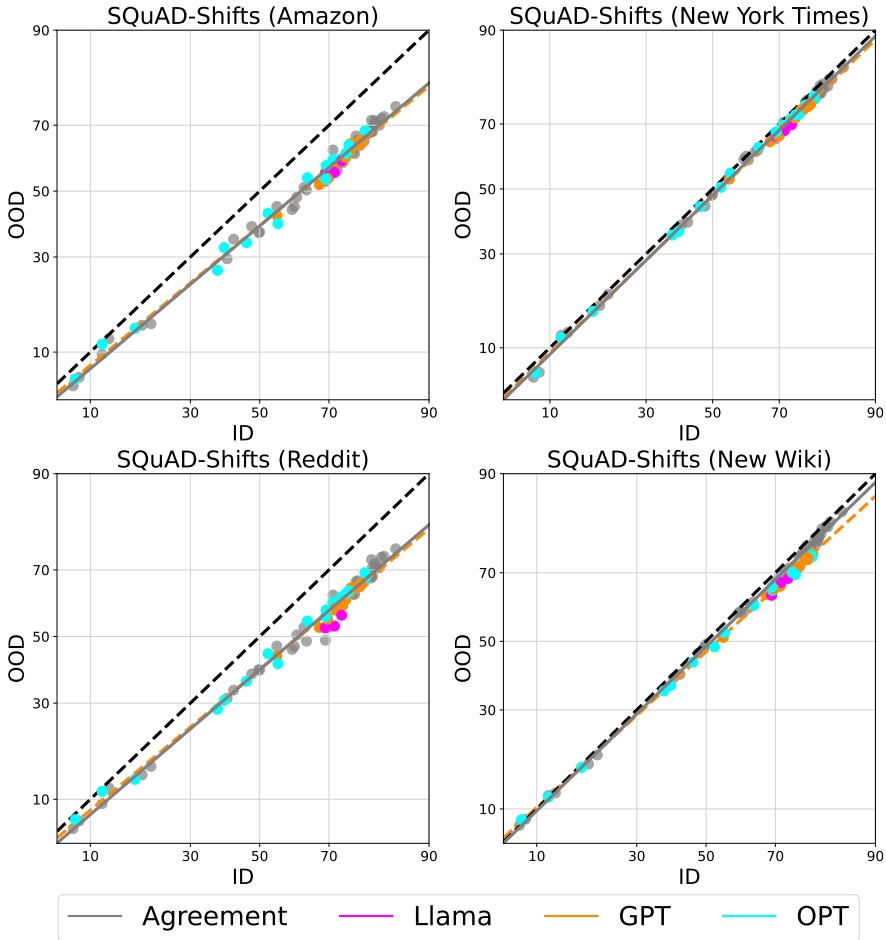

Figure 3: ACL and AGL observed on extractive Question Answering when computed over a set of models finetuned from different base foundation models. ACL and AGL are seen to hold for all four shifts of SQuAD-Shifts under this setup. The base models used here are OPT, GPT, and LLama.

different lines. We suspect that the pretraining datasets for the models in our study observe much more homogeneity. Second, the ID versus OOD agreement for pairs of models in this ensemble, including pairs of different base foundation models, retains a strong linear correlation and the slope and bias closely matches that of accuracy. As a result, different pretraining does not break AGL.

## 5 ESTIMATING OOD ACCURACY USING ALINE ON DIVERSE ENSEMBLES

With sufficient diversity in the ensemble, we observe that AGL succeeds over other OOD estimation baselines in terms of predicting the performance of the models in the ensemble. Specifically, we predict the performances of models in a collection consisting of 1) models trained from randomly-initialized heads (Section 3) and 2) different base models (Section 4). For image-classification, our model collection consist of just the former i.e. linear models over CLIP representations. Our question-answering model collection includes both i.e. GPT, OPT, Llama models individually trained from differently initialized heads. We compare the AGL prediction algorithms, ALine-S and ALine-D (Baek et al., 2022), with other existing methods: ATC (Garg et al., 2022), AC (Hendrycks & Gimpel, 2017b) and DOC-Feat (Guillory et al., 2021) that utilize model confidence to estimate OOD accuracy. We also assess the direct use of agreement to predict accuracy, dubbed naive agreement (Jiang et al., 2022; Madani et al., 2004). ALine-S simply transforms the ID accuracy using the slope and bias agreement estimates. ALine-D sets up and solves a system of $n$ choose 2 linear

equations where the OOD accuracy of each model are the variables. Empirically, Aline-D performs better than ALine-S. More details for these algorithms are provided in Appendix 9.2.

We observe that with the right diversity in the model collection (i.e. the ones exhibiting AGL), variants of the ALine algorithm surpass confidence/probability based methods by achieving the lowest error of predicting the OOD performance of fine-tuned foundation models on almost all tasks as seen in Table 3. For the confidence based methods (ATC, AC, DF), we pick the lower error value (from the ones obtained with and without the temperature scaling of the logits), even though in practice, this would not be known apriori. Though temperature scaling can be applied to calibrate models in terms of their accuracy, calibrating models for the F1 score by temperature scaling is not directly obvious. As a result, we observe that for extractive QA datasets, confidence based methods particularly suffer.

## 6 CONCLUSION

We develop methods for extending AGL to foundation models to enable OOD performance prediction in this emerging paradigm. We found that applying AGL directly may sometimes fail and to properly utilize this phenomena for performance estimation requires a careful tuning of the distribution of models in the ensemble for their errors to be uncorrelated. Unlike the original paradigm of AGL, where models observed tens or hundreds of epochs of training on the in-distribution dataset, we find that stochasticity in specific optimization choices, specifically random initialization, is crucial for foundation models. We also remark that our findings suggest that even large pretrained models with light fine-tuning could be very sensitive to corruptions in the representation learned, especially with a randomly initialized linear head. Second, though Baek et al. (2022) posed AGL as a model centric phenomena that is specifically only observed in neural network ensembles, we find that linear models could also observe AGL when the data and the distribution shift contain certain structures (as is possible in the CLIP representation space).

Our conclusion on AGL also sheds light on ACL (i.e. accuracy-on-the-line) in the presence of foundation models, a phenomenon that is of independent interest. Some recent works have studied the effect of different forms of fine-tuning on ACL (Radford et al., 2021; Awadalla et al., 2022). The main finding reported is that different forms of fine-tuning lead to different slopes in the linear correlations, a term that is often called "effective robustness". In our results, we find that when fine-tuned the same way, models obtained from *different base foundation models* all (OPT, GPT2, GPT2-Neo, and Llama2) lie on the *same* line (Figure 3). This is particularly intriguing because it goes against the common wisdom that the amount of pretraining data determines the effective robustness. We leave these questions for future analysis.

## 7 REPRODUCIBILITY STATEMENT

Appendix 9.1 outlines the hyperparameters we used to obtain our results, and Appendix 9.11 lists the Huggingface sources for each foundation model we evaluated. In order to make it easier to reproduce our findings, we plan to release code for linear probing, fine-tuning, and agreement calculation.

## 8 ETHICS STATEMENT

Estimating the out-of-distribution performance of foundation models has rapidly grown in importance, especially as these models are increasingly deployed in real-world use cases. Our work focuses on a promising method to measure OOD performance without using labeled data, which could be a valuable tool to identify when performance degrades due to distribution shift. This would enable deployers to reduce the harm of machine learning systems when they encounter OOD inputs. However, deployers should be careful to not use AGL as the only signal for OOD performance. The correlation between agreement and accuracy is not guaranteed to hold for all distribution shifts, so other metrics should additionally be used to monitor model performance. In particular for foundation models, we observe that careful choices during fine-tuning is required to observe AGL. Furthermore, while AGL may correctly predict that the average OOD performance remains high, it may not identify whether different subpopulations experience drastically different changes in performance. These subgroups could correspond to protected categories like gender and race, or to inputs where

wrong outputs are much more harmful than average. Future work could address this by carefully examining whether AGL can predict the effect of distribution shift on all subpopulations.

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

# 9    APPENDIX

## 9.1    FINETUNING SPECIFICS

We state here the specific parameters used in finetuning GPT2-Medium for extractive QA and CLIP for image classification. Across the four different sources of diversity, the epochs are varied regardless of the experiment. We train with AdamW as the optimizer (Loshchilov & Hutter, 2017). For randomly initializing linear heads we vary the seed for the head and keep all other values fixed. For changing the finetuning hyperparameters, we vary the learning rate and weight decay. To shuffle the data, we change the data seed that control the data ordering during training. And finally for data subsetting, we get different proportions of the dataset which are independently sampled.

For the GPT2-Medium models, we train a total of 50 models for studying the sources of diversity. For the CLIP models, we fine-tune upwards of 200 models (i.e. linear heads on top of the CLIP representation) for the different vision datasets.

| Source of Diversity | GPT2-Medium | |
|---|---|---|
| | Varied | Fixed |
| Random linear heads | LS: varied | LR: $3 \times 10^{-6}$ 
 WD: $2 \times 10^{-4}$ 
 DS: fixed 
 DP: 20% 
 EP: 0–3 
 B: 4 |
| Data shuffling | DS: varied | LR: $4 \times 10^{-6}$ 
 WD: $1 \times 10^{-4}$ 
 LS: fixed 
 DP: 10% 
 EP: 0–3 
 B: 4 |
| Data subsetting | DP: $4.5\% - 50\%$ | LR: $2 \times 10^{-6}$ 
 WD: $1 \times 10^{-4}$ 
 DS: varied 
 LS: fixed 
 EP: 1 
 B: 4 |

Table 4: Finetuning specifics for extractive QA (LR: learning rate, WD: weight decay, LS: linear head initialization seed, DS: data shuffling seed, DP: data subsetting proportion, EP: epochs, B: batch size)

| Source of Diversity | CLIP + ViT-B/32 (LAION-2B) | |
| --- | --- | --- |
| | **Varied** | **Fixed** |
| Random linear heads | LS: varied | LR: different per dataset
WD: 0
DS: fixed
DP: 100%
EP: 1–100
B: 1024 |
| Data shuffling | DS: varied | LR: different per dataset
WD: 0
LS: fixed
DP: 100%
EP: 1–100
B: 1024 |
| Data subsetting | DP: $10\% - 50\%$ | LR: different per dataset
WD: 0
DS: varied
LS: fixed
EP: 1–100
B: 1024 |

Table 5: Finetuning specifics for image classification (LR: learning rate, WD: weight decay, LS: linear head initialization seed, DS: data shuffling seed, DP: data subsetting proportion, EP: epochs, B: batch size)

## 9.2 ALINE-S/D

ALine is the OOD accuracy estimating metric that utilizes AGL (Baek et al., 2022), i.e. the close overlap between the ID vs OOD accuracy of models and agreement between pairs of models, over a set of models. There are two methods within ALine: ALine-S and ALine-D

Given $\text{Acc}_{\text{ID}}(f_1)$ and $\text{Agr}_{\text{OOD}}(f_1, f_2)$, when agreement holds, the relationship between the agreement line and accuracy line is as follows.

$$\Phi^{-1}(\text{Acc}_{\text{OOD}}(f_1)) = a \cdot \Phi^{-1}(\text{Acc}_{\text{ID}}(f_1)) + b \Leftrightarrow \Phi^{-1}(\text{Agr}_{\text{OOD}}(f_1, f_2)) = a \cdot \Phi^{-1}(\text{Agr}_{\text{ID}}(f_1, f_2)) + b \tag{2}$$

$\Phi^{-1}$ is the probit transform used to induce a better linear fit as used in Baek et al. (2022) and Miller et al. (2021). To find $Acc_{OOD}(f_2)$, we can estimate the slope $a$ and bias $b$ as follows and

$$\hat{a}, \hat{b} = \arg \min_{a,b \in \mathbb{R}} \sum_{i \neq j} \left( \Phi^{-1}(\hat{\text{Agr}}_{\text{OOD}}(h_i, h_j)) - a \cdot \Phi^{-1}(\hat{\text{Agr}}_{\text{ID}}(h_i, h_j)) - b \right)^2 \tag{3}$$

With $\hat{a}$ and $\hat{b}$, we can find $\text{Acc}_{\text{OOD}}(f_2)$ with the estimator for the ID accuracy $\text{Acc}_{\text{ID}}(f_2)$. This method is called Aline-S.

A similar method, ALine-D, uses pointwise accuracies and agreement of the model of interest instead of estimating the entire agreement line. If the models of interest are $h$ and $h'$, then the following holds.

$$\frac{1}{2} \left( \Phi^{-1}(\text{Acc}_{\text{OOD}}(h)) + \Phi^{-1}(\text{Acc}_{\text{OOD}}(h')) \right) = \frac{a}{2} \left( \Phi^{-1}(\text{Acc}_{\text{ID}}(h)) + \Phi^{-1}(\text{Acc}_{\text{ID}}(h')) \right) + \frac{b}{2} \tag{4}$$

With the fact that $b = \Phi^{-1}(\text{Agr}_{\text{OOD}}(h, h')) - a \cdot \Phi^{-1}(\text{Agr}_{\text{ID}}(h, h'))$, we have

$$\begin{aligned}
&\frac{1}{2} \left( \Phi^{-1}(\text{Acc}_{\text{OOD}}(h)) + \Phi^{-1}(\text{Acc}_{\text{OOD}}(h')) \right) \\
&= \Phi^{-1}(\text{Agr}_{\text{OOD}}(h, h')) + a \cdot \left( \frac{\Phi^{-1}(\text{Acc}_{\text{ID}}(h)) + \Phi^{-1}(\text{Acc}_{\text{ID}}(h'))}{2} - \Phi^{-1}(\text{Agr}_{\text{ID}}(h, h')) \right)
\end{aligned} \tag{5}$$

With the two unknowns, $\text{Acc}_{\text{OOD}}(h)$ and $\text{Acc}_{\text{OOD}}(h')$, and one equations we cannot find the unknowns. However, with more overlapping pairs, we can get the same number equations as variables and find the OOD accuracy of a model of interest.

## 9.3 EXPERIMENTING WITH SOURCES OF DIVERSITY FOR IMAGE CLASSIFICATION WITH CLIP

In Section 3.1, we examine the role of three separate sources of diversity that can be used to obtain a set of finetuned classifiers for image classification over the CIFAR10 dataset. Specifically, we consider training a linear probe on top of the CLIP features. In this section, we perform the same experiment, but with the Office-Home dataset (Venkateswara et al., 2017), a benchmark dataset consisting of 4 domains of images for 65 common objects: "Art", "Clip Art", "Product", and "Real". Furthermore, we also look at full fine-tuning for the CIFAR10 dataset; wherein a fully-connected linear layer is attached at the end of the CLIP model, and all of the weights in the model are trained using gradient steps over the Cross Entropy loss. We find, as noted previously, that for both these settings, using random linear heads at initialization leads to the desired diversity in the resultant model set for AGL behavior.

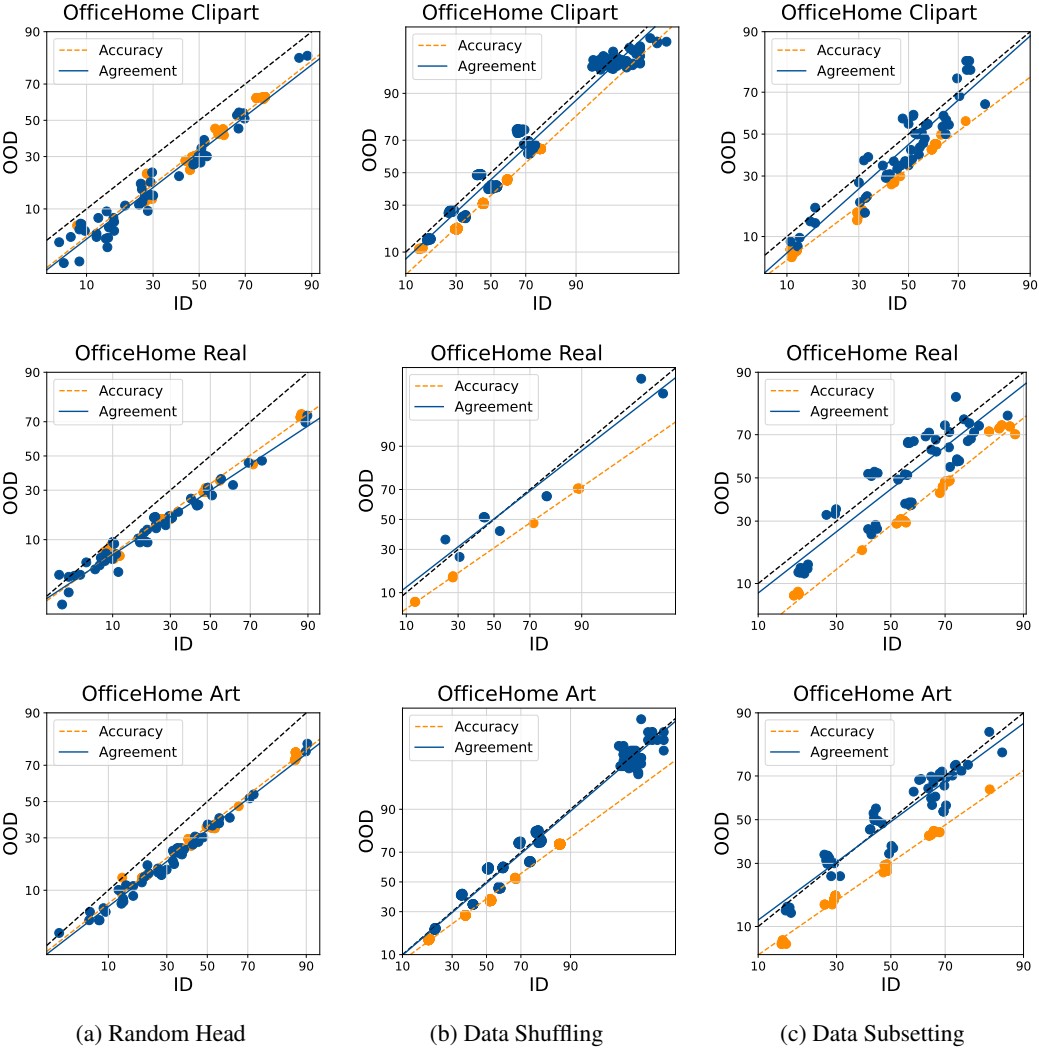

(a) Random Head      (b) Data Shuffling      (c) Data Subsetting

Figure 4: ID vs OOD trends of accuracy and agreement of linear probe finetuned CLIP models, when finetuned on the OfficeHome Art (top row), OfficeHome Product (middle row), OfficeHome Real (bottom row) datasets respectively. The titles of the graphs show the respective OOD dataset domain. Each column states the respective source of diversity used to obtain the model set. Clearly, using Random Head is the only method reliably yielding the diversity needed for AGL, and thus allowing accurate ALine OOD accuracy estimation.

| OfficeHome ID domain | Source of Diversity | OfficeHome OOD MAPE(%) | OfficeHome OOD MAE(%) |
|---|---|---|---|
| Art | Random Linear Heads | **15.43** | 5.38 |
| | Data Shuffling | 28.65 | 8.51 |
| | Data Subsetting | 35.08 | **4.38** |
| ClipArt | Random Linear Heads | **18.19** | 6.60 |
| | Data Shuffling | 30.64 | 8.28 |
| | Data Subsetting | 32.88 | **5.40** |
| Product | Random Linear Heads | **24.59** | **3.56** |
| | Data Shuffling | 222.18 | 25.68 |
| | Data Subsetting | 128.82 | 22.76 |
| Real | Random Linear Heads | **10.39** | **2.45** |
| | Data Shuffling | 44.55 | 12.42 |
| | Data Subsetting | 59.33 | 14.55 |

Table 6: The ALineD MAPE(%) and MAE(%) when applied to model sets obtained using the three sources of diversity, specifically by using one of the 4 domains (Art, ClipArt, Product, Real) of the OfficeHome dataset as ID (i.e. used for the finetuning) and the remaining three as OOD. As can be seen from Figure 4, the use of random linear heads at initialization is the only method consistently resulting in AGL behaviour, and the same can be seen in this table with the smallest MAPE and MAE values

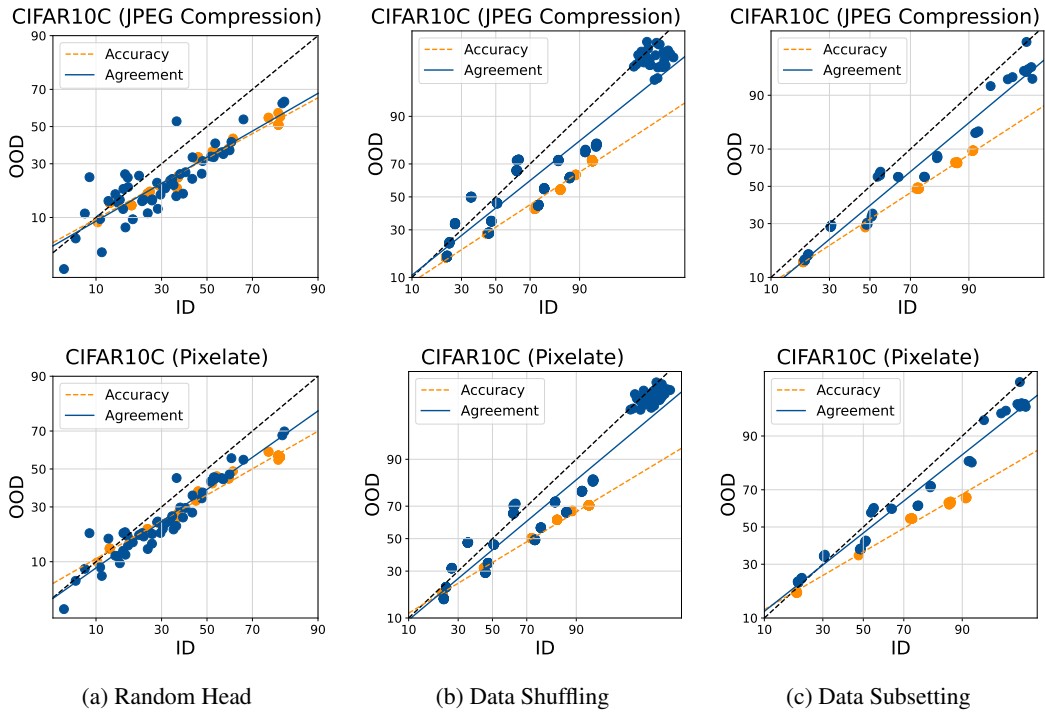

(a) Random Head      (b) Data Shuffling      (c) Data Subsetting

Figure 5: ID vs OOD trends of accuracy and agreement of fully fine-tuned CLIP models, when fine-tuned on the CIFAR10 dataset, and evaluated on the CIFAR10C "JPEG Compression" (top row) and "Pixelate" (bottom row) shifts. As observed with just training the linear probe atop CLIP, using Random Head is the only method reliably yielding the diversity needed for AGL, and thus allowing accurate ALine OOD estimation.

| Source of Diversity | CIFAR10C MAPE(%) | CIFAR10C MAE(%) |
|---|---|---|
| Random Linear Heads | **29.37** | **6.51** |
| Data Shuffling | 69.98 | 16.83 |
| Data Subsetting | 33.57 | 12.15 |

Table 7: The averaged MAPE(%) and MAE(%) for the ALine method when applied to estimate the OOD performance of fully fine-tuned CLIP models across all 19 CIFAR10C shifts. As seen in Fig 5, using Random Heads yields the closest AGL line, and thus the smallest ALine MAPE/MAE.

## 9.4 EXPERIMENTING WITH SOURCES OF DIVERSITY FOR EXTRACTIVE QUESTION ANSWERING WITH GPT

In Section 3.2 we look at the use of fully finetuned GPT2-Medium models for the task of extractive QA; specifically when finetuning on the SQuAD dataset, and evaluating on OOD data from the SQuAD-Shifts datasets. We consider the three sources of diversity for this finetuning i.e. using random linear heads, random ordering, and independent data subsetting. Figure 6 shows the results obtained in this setup.

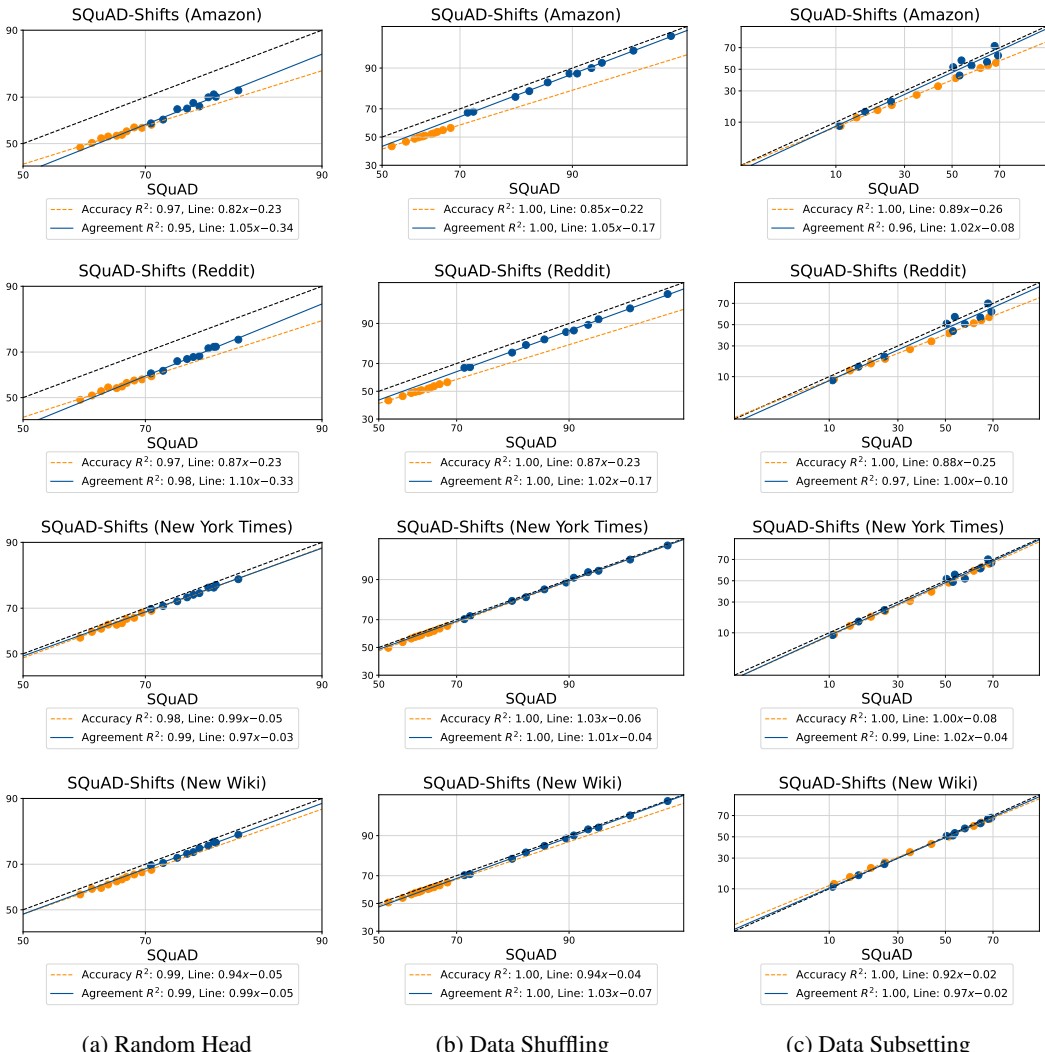

(a) Random Head          (b) Data Shuffling          (c) Data Subsetting

Figure 6: ID vs OOD trends of accuracy and agreement of LLMs finetuned for Question Answering from a single pretrained base model (GPT2-Medium). Each column presents trends for different sources of stochasticity employed to obtain a diverse ensemble of finetuned models. We see that random linear head initialization is the best method to obtain a set of models exhibiting AGL behavior

## 9.5 EXPERIMENTING WITH SOURCES OF DIVERSITY FOR EXTRACTIVE QUESTION ANSWERING WITH OPT AND BERT

In this section, we expand our evaluations to finetuned OPT-125M and BERT models for the extractive question answering task discussed in Section 3.2. For both of these base foundation models, we consider the three sources of diversity for finetuning i.e. using random linear heads, random ordering, and independent data subsetting, and plot the respective ID vs OOD accuracy of models and agreement between pairs of models in the resultant model set.

These experiments also afford us the chance to analyse the similarites and differences between the ACL/AGL trends exhibited by the model sets with GPT2-Medium, OPT-125M, and BERT as the base FM respectively. In particular, AGL is slightly worse for OPT-125M and BERT, and thus ALine has a higher error on OPT-125M and BERT than GPT2-Medium. However, we still see a consistent trend where AGL holds the best for random head initialization compared to data shuffling and data subsetting; thus implying that the ALine error for random head initialization is the smallest out of all diversity sources. Thus, the importance of random head initialization applies to all models regardless of architecture in AGL.

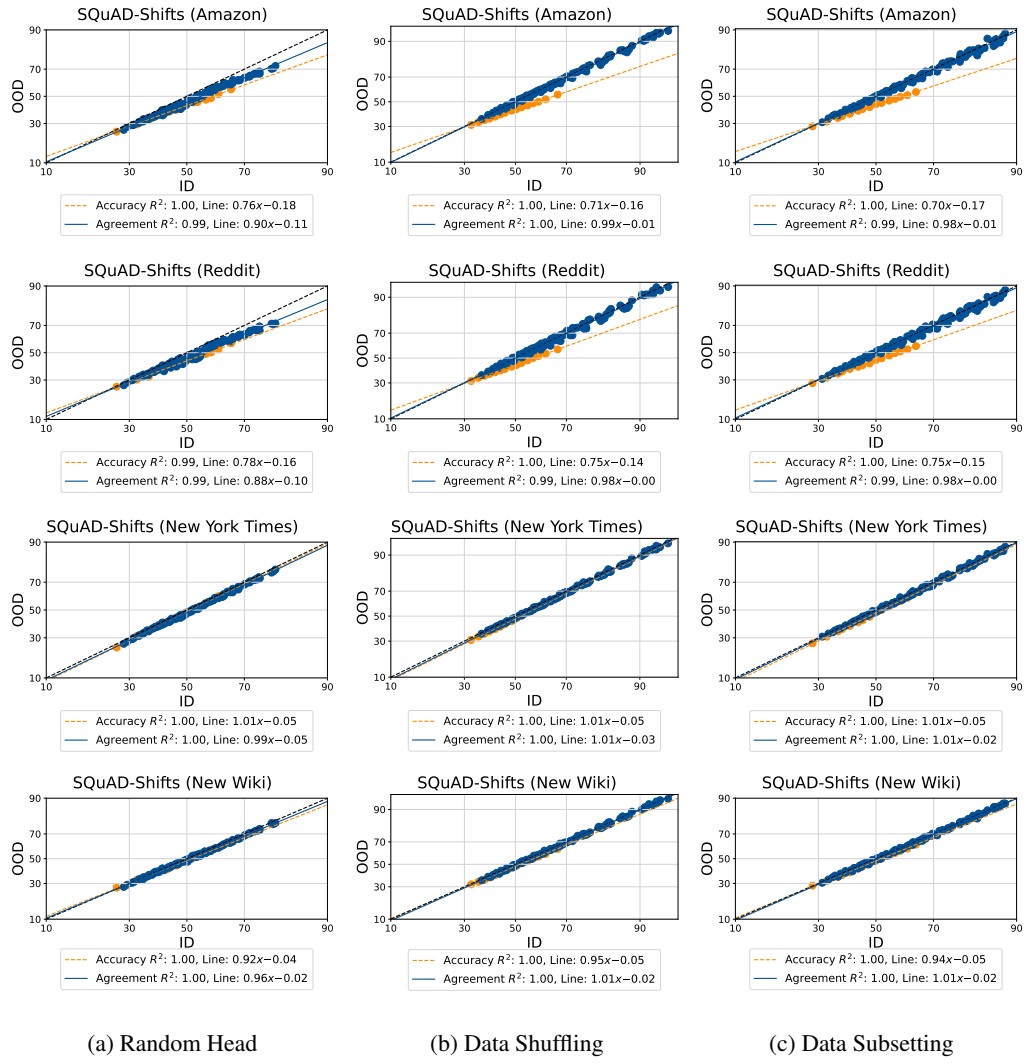

(a) Random Head  (b) Data Shuffling  (c) Data Subsetting

Figure 7: ID vs OOD trends of accuracy and agreement of LLMs finetuned for Question Answering from a single pretrained base model (OPT-125M). Similar to the GPT2-Medium results, these show that random linear head initialization is the best method to obtain model sets exhibiting AGL

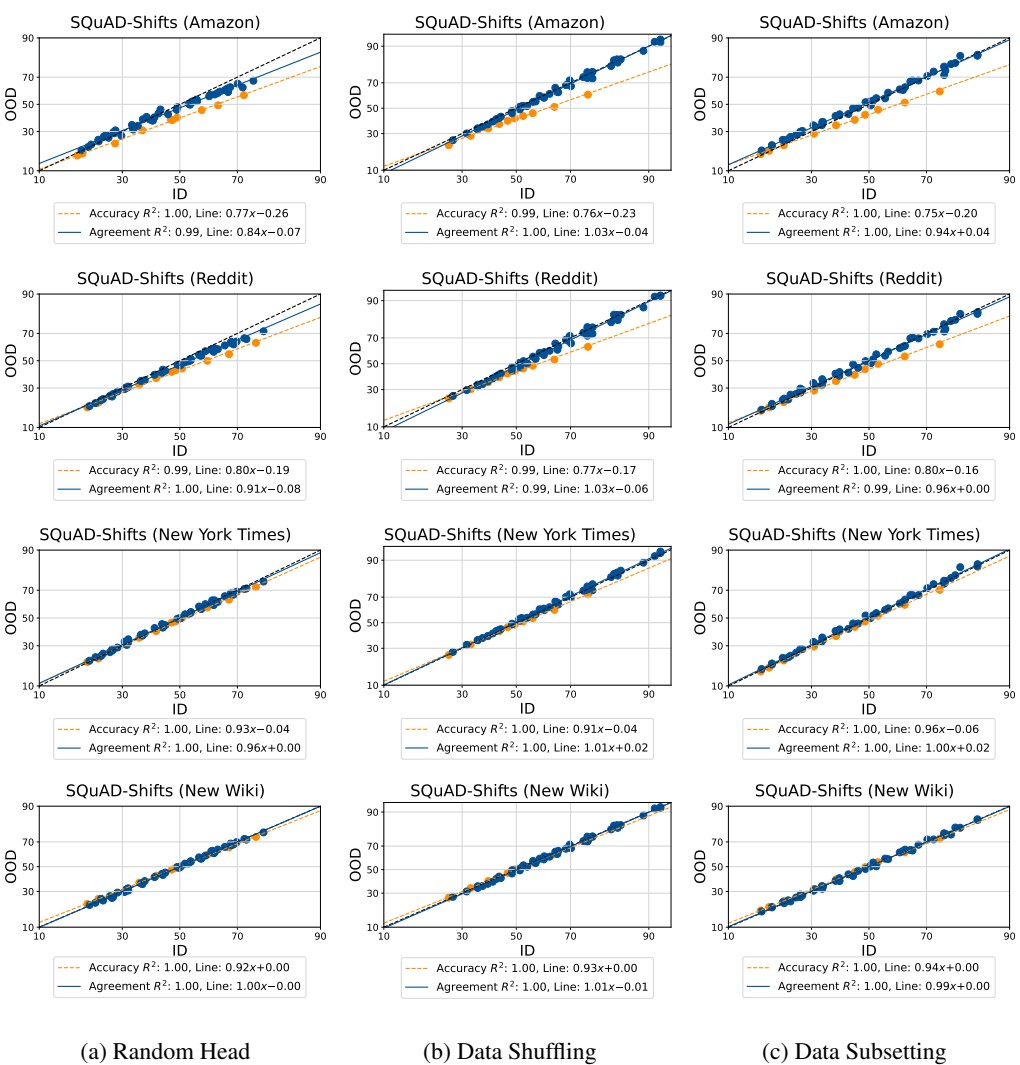

(a) Random Head    (b) Data Shuffling    (c) Data Subsetting

Figure 8: ID vs OOD trends of accuracy and agreement of LLMs finetuned for Question Answering from a single pretrained base model (BERT). Similar to the GPT2-Medium results, these show that random linear head initialization is the best method to obtain model sets exhibiting AGL

| Source of Diversity | SQuAD-Shifts Amazon MAPE (%) | SQuAD-Shifts Reddit MAPE (%) |
|---|---|---|
| Random Linear Heads | **6.54** | **5.43** |
| Data Shuffling | 11.37 | 8.70 |
| Data Subsetting | 11.15 | 9.65 |

Table 8: The averaged MAPE(%) for ALine-D when applied to estimate the OOD performance of OPT-125M fully fine-tuned model sets on SQuAD-Shifts Amazon and SQuAD-Shifts Reddit. As seen in Figure 7, using Random Heads yields the closest AGL line, and thus the smallest ALine-D MAPE for the respective model set.

| Source of Diversity | SQuAD-Shifts Amazon MAPE (%) | SQuAD-Shifts Reddit MAPE (%) |
|---|---|---|
| Random Linear Heads | **14.70** | **8.62** |
| Data Shuffling | 16.55 | 9.16 |
| Data Subsetting | 22.13 | 18.64 |

Table 9: The averaged MAPE(%) for ALine-D when applied to estimate the OOD performance of BERT fully fine-tuned model sets on SQuAD-Shifts Amazon and SQuAD-Shifts Reddit. As seen in Figure 8, using Random Heads yields the closest AGL line, and thus the smallest ALine-D MAPE for the respective model set.

### 9.6 EXPERIMENTING WITH SOURCES OF DIVERSITY FOR TEXT CLASSIFICATION WITH LLMS

As another extension to the experiments in section 3.2 for LLM-based text tasks we conduct additional experiments on text classification with linear probing for GPT2-Medium and OPT-125M with the ID dataset as MNLI and OOD dataset as SNLI. MNLI-matched (Wang et al., 2018) and SNLI (Bowman et al., 2015) are sentence pair classification datasets where the task is to determine the relationship (entailment, contradiction, and neutral) between two sentences. Similar to Section 3.2, we start with one model and fine-tune with random head initialization, data shuffling, and data subsetting. However, here we do linear probing instead of full fine-tuning. This was previously not feasible in question-answering due to poor performance. We train a GPT2-Medium and OPT-125M each.

Figure 9 shows the results for GPT2-Medium and Figure 10 shows the results for OPT-125M. Despite the limited spread in accuracy arising due to linear probing, we see consistently observe random head initialization to be the most predictive setting for ALine. ALine achieve the least MAPE with random head initialization using OPT-125M (5.4%) and a low error similar to data subsetting with GPT2-Medium (5.9%). Complete results can be seen in Table 10. For both GPT2-Medium and OPT-125M, the agreement and accuracy lines match the closest for random head initialization.

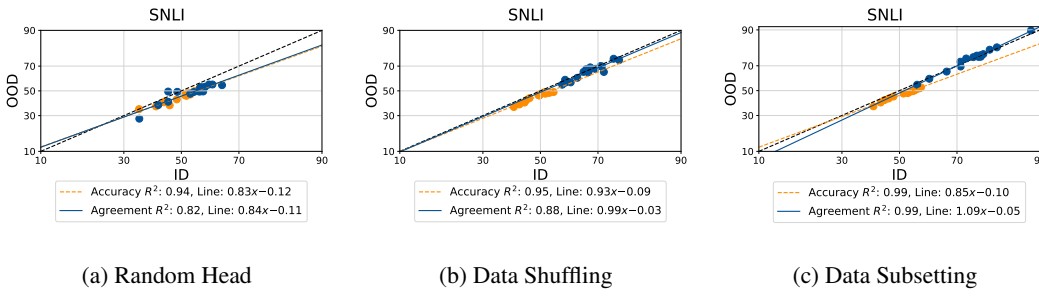

(a) Random Head        (b) Data Shuffling        (c) Data Subsetting

Figure 9: ID vs OOD trends of accuracy and agreement of LLMs finetuned for Text Classification from a single pretrained model (GPT2-Medium) on SNLI. Each column presents trends for different sources of diversity. These show that random linear initialization of the head exhibit the best AGL behavior

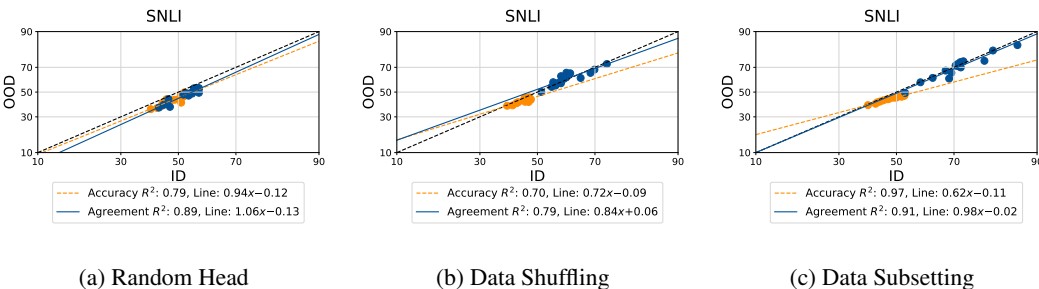

(a) Random Head        (b) Data Shuffling        (c) Data Subsetting

Figure 10: ID vs OOD trends of accuracy and agreement of LLMs finetuned for Text Classification from a single pretrained model (OPT-125M) on SNLI. Each column presents trends for different sources of diversity. These show that random linear initialization of the head exhibit the best AGL behavior

| Model | GPT2-Medium | OPT-125M |
|---|---|---|
| Random Linear Heads | 5.9 | **5.4** |
| Data Shuffling | 6.7 | 12.4 |
| Data Subsetting | **5.3** | 8.6 |

Table 10: The averaged MAPE(%) for ALine-D for text classiciation (MNLI-matched v.s. SNLI) on GPT2-Medium and OPT-125M

### 9.7 STUDYING AGL FOR TEXT CLASSIFICATION IN ENSEMBLES OF MULTIPLE BASE FOUNDATION MODELS

As an extension to section 4, we gather multiple base models for text classification and show AGL. The ID dataset is MNLI-matched and the OOD datasets are MNLI-mismatched, SNLI, HANS, and WNLI. MNLI/WNLI (Wang et al., 2018), SNLI (Bowman et al., 2015), and HANS (McCoy et al., 2019) are sentence pair classification datasets where the task is to determine the relationship (entailment, contradiction, and neutral) between two sentences. We fully fine-tune the base models of GPT, OPT, and Llama.

Specifically, on MNLI-mismatched and SNLI, AGL and ACL both hold. On HANS and WNLI, we can see that there is a low correlation in the accuracy and agreement. Thus, AGL does not hold. These results verify that the phenomenon of AGL only applies to datasets where the correlation for AGL and ACL are both high, as mentioned in Baek et al. (2022).

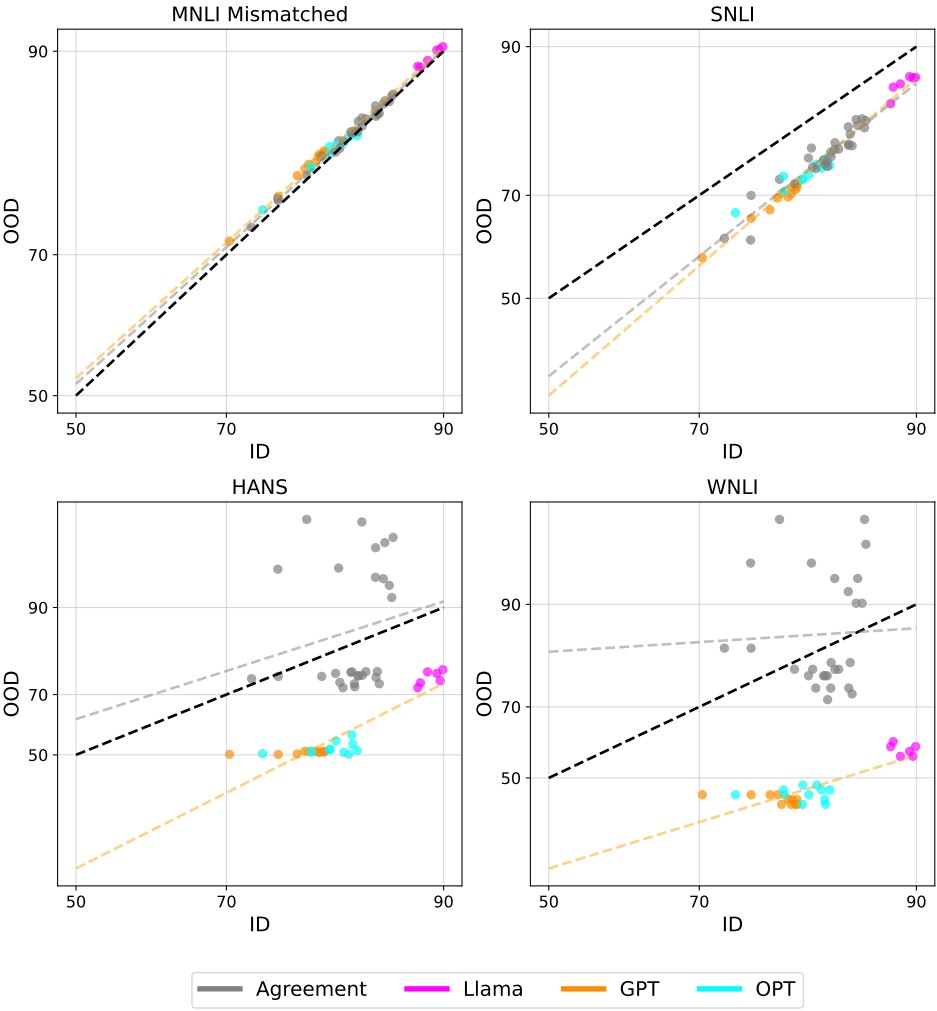

Figure 11: ID vs OOD trends of accuracy and agreement on text classification when computed over a set of models finetuned from different base foundation models. The shifts presented here are - MNLI-matched v.s. MNLI-mismatched/SNLI/HANS/WNLI. We see that ACL and AGL hold for MNLI-mismatched and SNLI. On HANS and WNLI, ACL does not hold and consequently AGL doesn't hold either.

### 9.8 Using Random-Head initialized fine-tuned CLIP models for other datasets

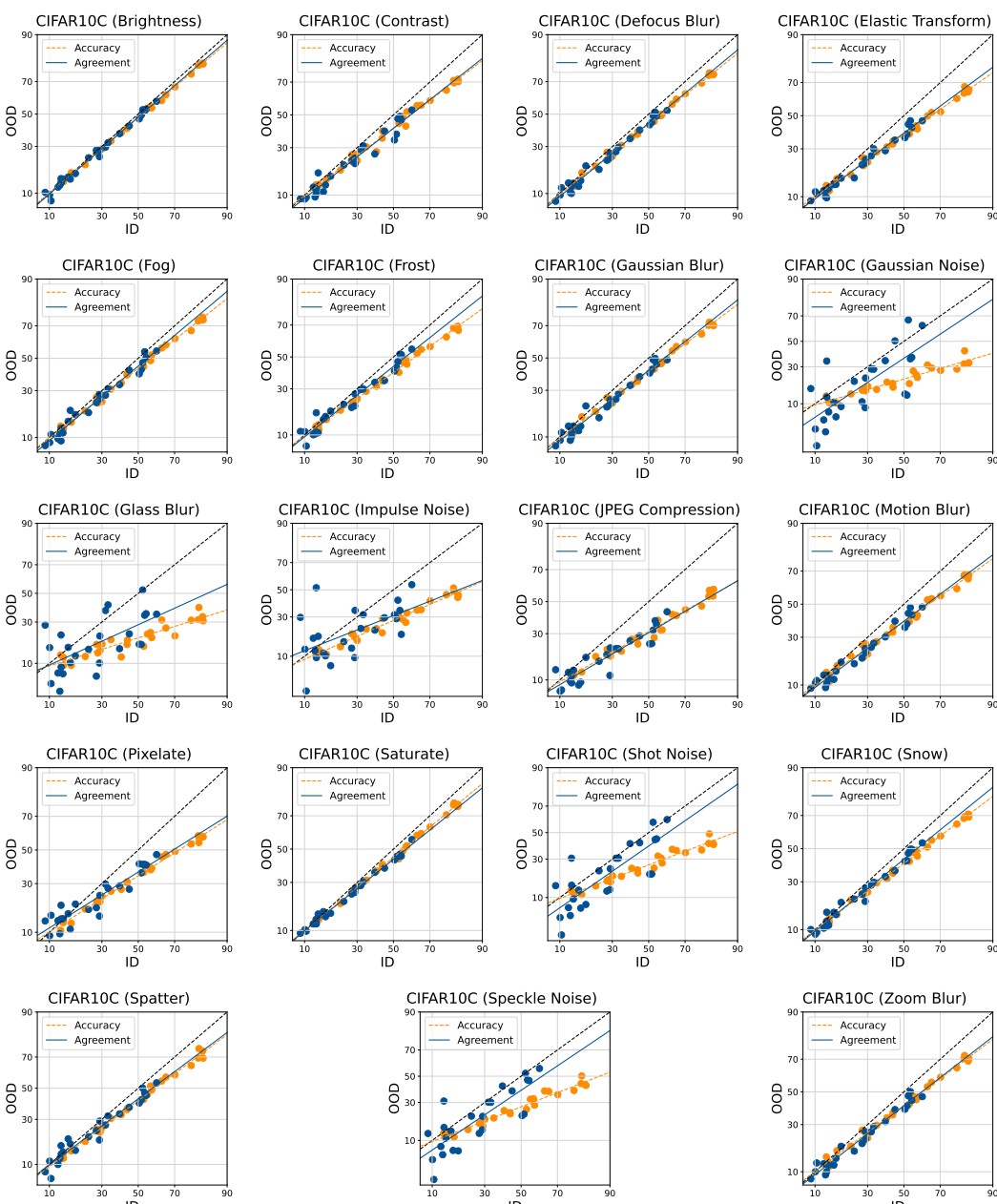

Figure 12: AGL and ACL for all C10C shifts with random head initialization fine-tuning.

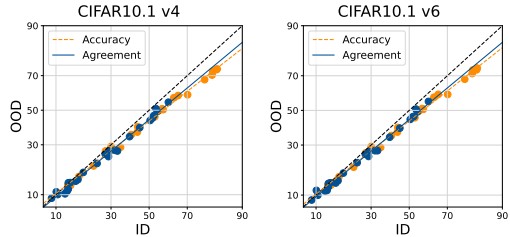

Figure 13: AGL and ACL for the C10.1 shifts with random head initialization fine-tuning.

Figure 14: AGL and ACL for the C100C shifts with random head initialization fine-tuning.

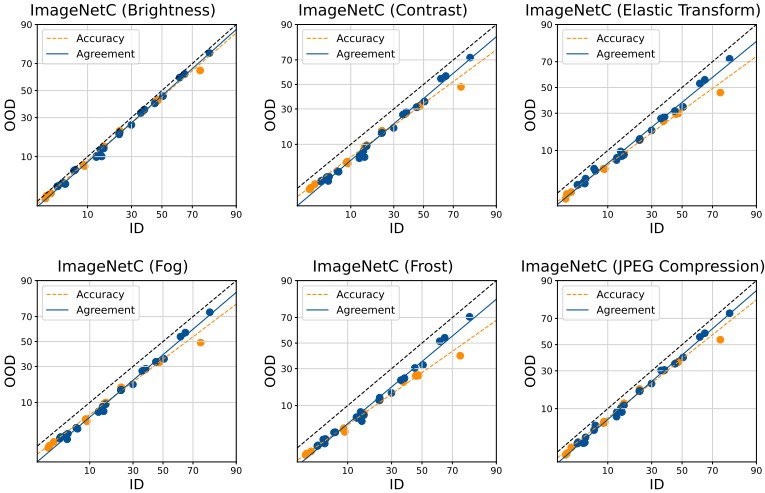

Figure 15: AGL and ACL for the ImageNetC shifts with random head initialization fine-tuning.

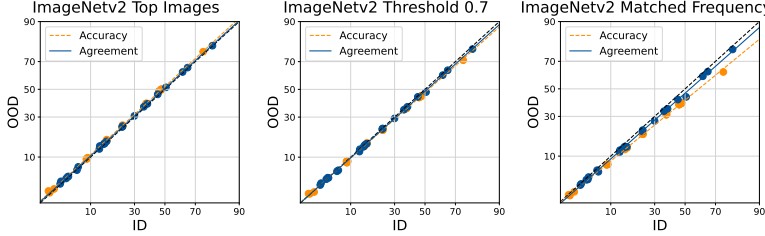

Figure 16: AGL and ACL for the ImageNet V2 shifts with random head initialization fine-tuning.

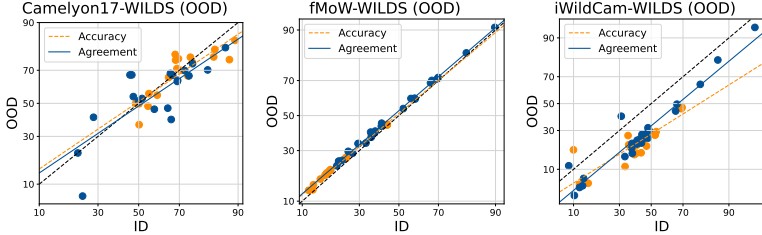

Figure 17: AGL and ACL for 3 benchmarks from the WILDS dataset with random head initialization fine-tuning.

## 9.9 ANALYSING ALINE PREDICTIONS VS OOD ACCURACIES

This section presents another viewpoint of AGL when the base models are pretrained on different dataset are finetuned. Figure 18 presents plots of how well ALineD and its competing baselines predict OOD accuracy across a large spectrum of ground truth OOD accuracies. It can be noted that ALineD shows a strong linear correlation described by a line that almost coincides with the diagonal $y = x$. In comparison, other baselines are often off diagonal and show weaker correlation between prediction and ground truth. Consequently, it can be seen that ALineD predicts the OOD accuracy faithfully across the entire range accuracy values while doing much better than competing baselines.

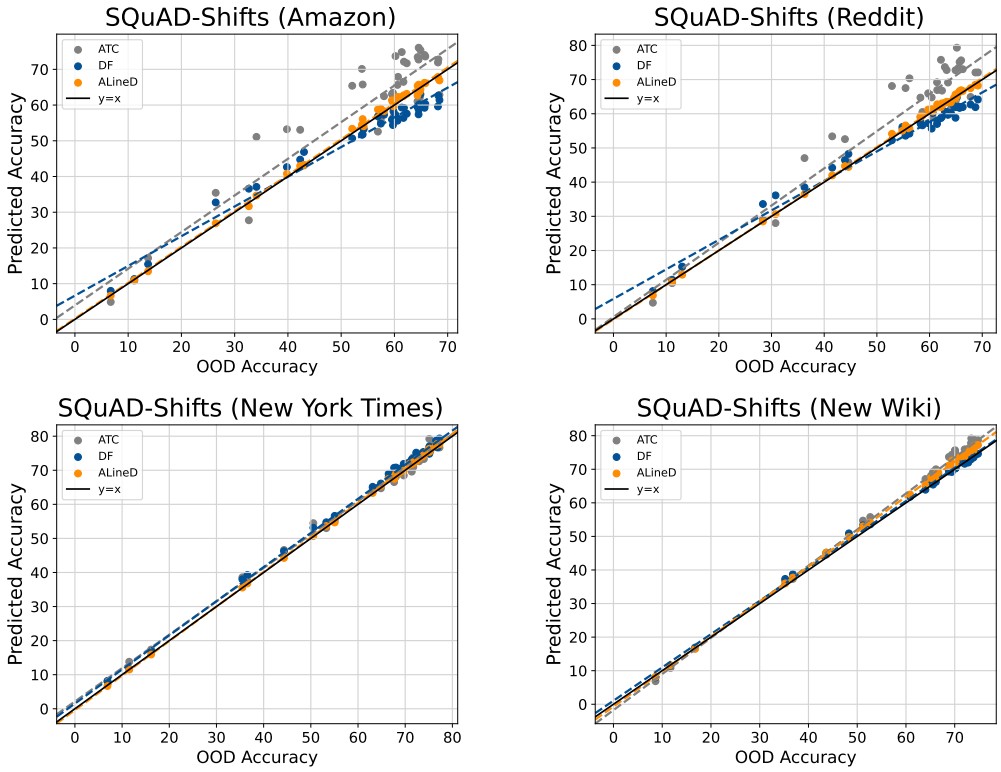

Figure 18: OOD Accuracy vs Predicted Accuracy. This figure shows the performance estimation for a wide range of true accuracy values for various methods. The setting shown is one with models finetuned from pretrained models with varying pretraining data.

## 9.10   ANALYSIS OF PREDICTION CORRELATION

| Method | SQuAD-Shifts Amazon | SQuAD-Shifts Reddit |
|--------|---------------------|---------------------|
| ALine-D | **0.98** | **0.98** |
| ProjNorm | 0.64 | 0.79 |

Table 11: Correlation coefficient between OOD Accuracy and Prediction for ALine-D and ProjNorm

In this section we present a comparison with ProjNorm Yu et al. (2022), a method that yields a score which is shown to be correlated with the OOD performance of the model. We study the same setting as presented in Section 4 where we estimate OOD performance of foundation models pretrained on different text corpora. Unlike AGL, ProjNorm doesn't provide an estimate of the OOD performance hence we compare the linear correlation between the predicted value and OOD performance. From Table 11 it can be seen that estimates from ALine-D are more strongly correlated with OOD performance than ProjNorm. We found that some models (OPT-125M) showed almost no linear correlation with ProjNorm and removed those before computing the correlation coefficient so as to have a comparison in a setting where ProjNorm has reasonable performance.

## 9.11 HUGGINGFACE LINKS

Here are the Huggingface links to the pretrained base foundation models we finetuned: GPT2 (https://huggingface.co/gpt2), GPT2-Medium (https://huggingface.co/gpt2-medium), GPT2-Large (https://huggingface.co/gpt2-large), GPT2-XL (https://huggingface.co/gpt2-xl), GPT-Neo-125M (https://huggingface.co/EleutherAI/gpt-neo-125m), GPT-Neo-1.3B (https://huggingface.co/EleutherAI/gpt-neo-1.3B), OPT-125M (https://huggingface.co/facebook/opt-125m), OPT-1.3B (https://huggingface.co/facebook/opt-1.3b), Llama2-7B (https://huggingface.co/meta-llama/Llama-2-7b-hf), Alpaca-7B (https://huggingface.co/WeOpenML/Alpaca-7B-v1), Vicuna-7B (https://huggingface.co/lmsys/vicuna-7b-v1.3), BERT (https://huggingface.co/bert-base-uncased)

## 9.12 ERROR OF OOD PREDICTIONS AT DIFFERENT ACCURACY VALUES

Table 12 presents a breakdown of prediction MAE at different true OOD performance levels. Results presented here are for sets of models finetuned from different base foundation models. We find that ALine achieves low MAE across the accuracy spectrum, i.e., ALine is predictive of model performance irrespective of whether the true OOD performance is high or low. For practitioners interested in knowing their model's performance OOD at the end of training, the low MAE at higher accuracy ranges indicates that ALine can be reliably used in such situations.

| OOD Dataset | Accuracy Range | ALine-D | ALine-S | Naive Agr | ATC | AC | DF |
|---|---|---|---|---|---|---|---|
| SQuAD-Shifts Reddit | 0-30 | 0.28 | 1.08 | - | 19.08 | 83.23 | 11.88 |
| | 30-50 | 0.35 | 1.46 | 35.22 | 17.31 | 69.03 | 10.40 |
| | 50-70 | 0.62 | 0.99 | 15.54 | 10.70 | 41.27 | 4.82 |
| | 70-100 | - | - | - | - | - | - |
| SQuAD-Shifts Amazon | 0-30 | 0.24 | 1.14 | - | 25.19 | 84.09 | 14.05 |
| | 30-50 | 0.68 | 2.29 | 35.54 | 29.19 | 70.68 | 11.07 |
| | 50-70 | 0.98 | 1.24 | 15.69 | 11.77 | 43.23 | 6.27 |
| | 70-100 | - | - | - | - | - | - |
| SQuAD-Shifts Nyt | 0-30 | 0.16 | 0.51 | - | 14.14 | 83.39 | 8.38 |
| | 30-50 | 0.12 | 0.42 | 57.36 | 7.52 | 65.52 | 6.28 |
| | 50-70 | 0.45 | 0.72 | 13.90 | 1.76 | 39.55 | 3.28 |
| | 70-100 | 0.61 | 0.69 | 13.29 | 1.61 | 27.89 | 1.95 |
| SQuAD-Shifts New Wiki | 0-30 | 0.25 | 0.31 | - | 11.20 | 86.31 | 3.06 |
| | 30-50 | 1.18 | 1.41 | 48.56 | 2.92 | 67.24 | 5.02 |
| | 50-70 | 2.04 | 2.12 | 14.37 | 5.14 | 37.93 | 1.38 |
| | 70-100 | 2.32 | 2.22 | - | 5.31 | 29.94 | 0.49 |

Table 12: MAE (%) of predicting OOD performance at different true OOD accuracy ranges. Empty cells signify the absence of any suitable models in the desired range to compute predictions with the associated method.

| ID Dataset | OOD Dataset | Accuracy Range | ALine-D | ALine-S | Naive Agr | ATC | AC | DF |
|---|---|---|---|---|---|---|---|---|
| CIFAR10 | CIFAR10-C | 0-25 | 1.32 | 1.43 | 12.70 | 4.60 | 6.17 | 5.93 |
| | | 25-50 | 2.29 | 2.27 | 13.77 | 8.02 | 7.96 | 10.42 |
| | | 50-75 | 1.96 | 2.30 | 18.97 | 9.97 | 4.32 | 7.88 |
| | | 75-100 | 0.91 | 1.28 | - | 22.00 | 2.01 | 1.04 |
| CIFAR10 | CIFAR10.1 | 0-25 | 0.26 | 0.36 | - | 2.09 | 1.76 | 1.10 |
| | | 25-50 | 0.77 | 1.18 | 15.91 | 10.44 | 1.76 | 3.31 |
| | | 50-75 | 1.71 | 1.74 | 16.75 | 10.22 | 3.05 | 5.08 |
| | | 75-100 | - | - | - | - | - | - |
| CIFAR100 | CIFAR100-C | 0-25 | 1.34 | 1.60 | 13.41 | 10.62 | 2.52 | 4.94 |
| | | 25-50 | 4.70 | 4.62 | 11.96 | 15.65 | 4.11 | 10.53 |
| | | 50-75 | 4.35 | 3.98 | 16.51 | 20.91 | 2.83 | 7.39 |
| | | 75-100 | - | - | - | - | - | - |
| ImageNet | ImageNet-C | 0-25 | 1.05 | 0.91 | 9.29 | 8.88 | 1.57 | 2.55 |
| | | 25-50 | 6.23 | 6.07 | 16.27 | 19.33 | 4.77 | 6.47 |
| | | 50-75 | 10.98 | 11.07 | - | 9.63 | 1.82 | 7.47 |
| | | 75-100 | - | - | - | - | - | - |
| ImageNet | ImageNet V2 | 0-25 | 0.47 | 0.50 | 8.50 | 11.59 | 1.02 | 1.00 |
| | | 25-50 | 2.24 | 2.20 | 17.46 | 24.85 | 2.11 | 2.41 |
| | | 50-75 | 3.67 | 4.30 | 23.50 | 10.88 | 1.67 | 1.24 |
| | | 75-100 | 0.69 | 0.84 | - | 2.65 | 1.41 | 1.71 |
| FMoW | FMoW (val) | 0-25 | 0.60 | 0.73 | 18.21 | 11.53 | 1.93 | 1.90 |
| | | 25-50 | 0.82 | 0.76 | 24.24 | 10.19 | 2.06 | 0.95 |
| | | 50-75 | - | - | - | - | - | - |
| | | 75-100 | - | - | - | - | - | - |
| iWildCam | iWildCam (val) | 0-25 | 3.73 | 5.41 | 8.58 | 14.49 | 7.00 | 13.14 |
| | | 25-50 | 6.49 | 4.41 | 14.90 | 10.04 | 3.68 | 7.91 |
| | | 50-75 | - | - | - | - | - | - |
| | | 75-100 | - | - | - | - | - | - |
| Camelyon | Camelyon (val) | 0-25 | - | - | - | - | - | - |
| | | 25-50 | 5.60 | 6.00 | - | 12.02 | 12.94 | 3.41 |
| | | 50-75 | 6.19 | 4.60 | 12.20 | 7.50 | 12.43 | 2.73 |
| | | 75-100 | 7.08 | 9.26 | - | 7.42 | 9.07 | 6.30 |
| OfficeHome Art | other OfficeHome domains | 0-25 | 1.42 | 2.21 | 15.98 | 9.96 | 4.40 | 4.50 |
| | | 25-50 | 1.90 | 2.35 | 12.47 | 12.65 | 4.60 | 2.65 |
| | | 50-75 | 3.99 | 5.57 | 18.15 | 19.98 | 4.29 | 3.94 |
| | | 75-100 | 5.84 | 7.74 | - | 16.75 | 3.15 | 1.29 |
| OfficeHome ClipArt | other OfficeHome domains | 0-25 | 1.78 | 1.85 | 18.08 | 9.62 | 2.54 | 3.12 |
| | | 25-50 | 5.17 | 3.92 | 14.63 | 10.77 | 4.20 | 2.36 |
| | | 50-75 | 9.47 | 10.48 | 18.72 | 10.54 | 8.11 | 2.17 |
| | | 75-100 | 9.42 | 12.16 | - | 12.39 | 3.59 | 1.98 |
| OfficeHome Product | other OfficeHome domains | 0-25 | 1.23 | 1.34 | 13.18 | 8.30 | 8.91 | 11.99 |
| | | 25-50 | 2.15 | 2.53 | 15.30 | 4.97 | 5.07 | 16.60 |
| | | 50-75 | 8.00 | 10.90 | 19.59 | 9.26 | 5.33 | 5.90 |
| | | 75-100 | - | - | - | - | - | - |
| OfficeHome Real | other OfficeHome domains | 0-25 | 1.15 | 1.96 | 15.67 | 13.03 | 5.20 | 6.27 |
| | | 25-50 | 1.53 | 2.06 | 15.85 | 5.44 | 3.44 | 6.10 |
| | | 50-75 | 1.90 | 2.01 | 17.13 | 9.22 | 3.67 | 6.30 |
| | | 75-100 | 1.54 | 2.70 | - | 9.07 | 2.18 | 2.61 |

Table 13: MAE (%) of predicting OOD performance at different true OOD accuracy ranges. Empty cells signify the absence of any suitable models in the desired OOD accuracy range with the associated method.

