# OpenReview forum: "Predicting the Performance of Foundation Models via Agreement-on-the-line"
_ICLR.cc/2024/Conference — Submitted to ICLR 2024_

### Official Review · Reviewer_sUjj · 2023-10-31

**Soundness:** 3 good
**Presentation:** 3 good
**Contribution:** 3 good
**Rating:** 5
**Confidence:** 4

**Summary:**

This work explores how to accurately predict the out-of-distribution performance of foundation models in the case where one does not have access to labels for the OOD data. The authors rely on an intriguing observation made in earlier works: In order to predict OOD performance, it often suffices to determine to what degree different models agree on the in-distribution data, compared to the OOD data. While this phenomenon has been established for models trained from scratch, the strategy has not been extended to the case of pre-trained models. Such an extension comes with challenges: how can one ensure model diversity if the same foundation model is fine-tuned multiple times? The authors explore multiple methods to ensure diversity for linear probing for CLIP and demonstrate that accurate OOD prediction can be achieved if the linear heads are randomly initialised. The authors go one step further and demonstrate that reliable OOD prediction can even be achieved when employing several foundation models, that all are pre-trained on different data.

**Strengths:**

1. Practitioners these days largely rely on fine-tuning foundation models instead of training models from scratch. Thus extending the techniques of Baek et al. to cover the case of fine-tuning is very valuable and might prove very useful to practice. I thus think the results in this work are a timely contribution.
2. I think understanding how to induce diversity when dealing with a single foundation model is an interesting question on its own, especially when thinking of building ensembles based on multiple such training runs. The findings presented in this work are very surprising with that regard, e.g. using different initialisations for linear heads seems to play a very important role, which goes against my personal intuition.
3. The method seems to work pretty well empirically across different modalities, if things are tuned correctly, which is very encouraging and makes the contributions more relevant also to practically-oriented people.

**Weaknesses:**

1. I think the authors use the word **diversity** without explaining too much what exactly they mean and what is needed in order to make the agreement-on-the-line method work. Clearly there needs to be some diversity (otherwise all models would agree everywhere as correctly pointed out by the authors) but can there be too much diversity? This is especially confusing because the authors find that introducing stochasticity solely by using different initialisations is actually the most performant choice. But clearly, this cannot be more diverse than further using different batch orderings or even subsampling the data? Linear probing should even be a convex problem, so all runs should actually converge to the same minimum even from different initialisations, given that optimisation is performed for long enough. Could you elaborate on the role of diversity, e.g. can there be too much diversity? I think explaining this better would really help the subsequent empirical exploration.
2. In general, many empirical phenomena are observed but the authors do not really make an attempt at explaining them. Why does only using differently initialised random heads give the “right” amount of diversity? Why do you observe a way higher rate of agreement OOD compared to ID when using other techniques such as data shuffling and subletting etc? If anything, I would have expected more agreement in-distribution as all the models at least were optimised for this. What do you mean by strictly lying on the diagonal line y=x? Wouldn’t that suggest that in-distribution and out-of-distribution agreement are of the same magnitude?  What happens if instead of linear probing, you perform full fine-tuning in case of CLIP? Does the additional diversity also hurt the predictive performance?
3. It remains a bit unclear to me to what degree this method needs to be first validated before the results can be trusted. At least for linear probing with CLIP, getting the amount of diversity right seems to be very tricky as the method is highly unstable to small deviations. Moreover, in almost all cases, the experiments still show agreement-on-the-line, but don’t necessarily correlate with test accuracy (i.e. sharing the same slope and intercept). Thus observing agreement-on-the-line does not suffice to conclude that extrapolated accuracy values will actually be accurate. I would appreciate if the authors could discuss more how their observations regarding diversity relate to the prior work Baek et al. Is the method significantly more stable to changes in the training protocol when training from scratch?

**Questions:**

1. What are the units of the x-axis and y-axis for Figure 1 and Figure 2? How is agreement measured? Is this a log-log scale? How are the linear fits obtained?
2. What are the absolute test values in Table 1 and Table 2? Does the method tend to over-estimate or under-estimate the true test accuracy? It’s also not clear how significant a deviation of 5% is if the absolute values are not known. E.g. if test performance is 20%, then a deviation of 5% is clearly more significant than if test performance is 95%.

---

> ### Author Response · Authors · 2023-11-17
> **Thank you for your time and constructive feedback**
>
> Dear Reviewer,
>
> Thank you very much for your time and thorough feedback! We have now made the following changes in response to your feedback:
> 1. To illustrate the significance of the random head initialization for obtaining ensembles exhibiting Agreement-on-the-line (AGL)
>     - We study CLIP full fine-tuning with CIFAR10 on CIFAR10-C Shifts (see [Table 7](https://i.imgur.com/RmhQc3O.png))
>     - We extend the results of CLIP linear probe fine-tuning to the Office-Home dataset (see [Table 6](https://i.imgur.com/eYkJbL5.png))
>     - We train OPT and BERT models with linear probe fine-tuning for Question Answering (see [Table 8 and 9](https://i.imgur.com/qMePdYh.png))
>     - We add a new Text Classification study (with both linear probe, and full fine-tuning) using GPT, OPT, and LLaMa (see [Table 10](https://i.imgur.com/0bKxH0V.png))
> 2. We have changed the metric from MAE (Mean Absolute Error) to MAPE (Mean Absolute Percentage Error) to show the significance of the OOD accuracy estimates, as the MAPE captures the relative not just absolute deviation from the true value. We have also added a table with the MAE for different OOD accuracy ranges ([Appendix Section 9.12](https://i.imgur.com/5bzYHUb.png)) to show the consistent performance of ALine across high- and low-accuracy models.
> 3. We have provided some preliminary intuition as to the specific advantage of random linear head initialization for obtaining ensembles which exhibit AGL in the response below. We can add this in our paper if it would be helpful.
> We address your concerns below in more detail.
>
> ### **[Use of the word diversity] I think the authors use the word diversity without explaining too much what exactly they mean and what is needed in order to make the agreement-on-the-line method work. Clearly there needs to be some diversity (otherwise all models would agree everywhere as correctly pointed out by the authors) but can there be too much diversity?**
> Thank you for the feedback! We apologize for the confusion caused by our overloaded use of the term diversity.
> 1. At a high level, like you rightly said, we need some diversity in model predictions, else the models will agree too much. However, not just any kind of diversity is sufficient—for example, just adding noise to the models’ predictions would make them diverse, but not in a way that would help us use agreement to predict OOD accuracy. It is not just an overall measure of diversity that matters, but the “type” or “form” of diversity.
> 2. This paper shows that random head initialization leads to the right kind of diversity in the predictions of lightly fine-tuned models; thus causing the agreement and accuracy lines to have the same slope and intercept i.e. Agreement-on-the-line (AGL). Other sources of diversity lead to just ACL (i.e. strong linear correlation between the ID and OOD accuracies).
> 3. We will make this clear in our revised draft, that the big challenge in using foundation models is introducing some diversity, as just having any diverse ensemble on its own is not sufficient.
> 4. If we use models that are trained on drastically different training sets, they might be “too diverse”, in that the accuracies of these models might follow completely different linear trends between ID and OOD accuracy (i.e. ACL itself might not hold). We clarify again that we do not claim to have a precise measure of diversity, and even more so, AGL requires the right type of diversity.
>
> ### **[Conceptual understanding] Why does only using differently initialized random heads give the “right” amount of diversity?**
> This is indeed a surprising observation, and we think it would be interesting future work to understand this precisely. In particular, we find this observation to be true even for linear probing on CLIP features, which is potentially amenable to theoretical analysis. Our (admittedly imprecise) intuition is that since we are doing only a few steps of fine-tuning, we never move too far from the random head initialization. Hence with different initializations, we end up with models that can be quite “far” from each other, despite starting from the same foundation model. This paper [3] shows that random initialization dictates how the pretrained features end up changing, and provides some insight into why the head initialization can be so important. On the other hand, if we fix the random initialization, we might not end up moving too far, and as a result even with different data subsetting or data ordering, we might end up with very similar models that agree a lot ID and OOD, i.e. causing the agreement line to lie much higher than the accuracy line.

---

> > ### Author Response · Authors · 2023-11-17
> > **Rebuttal Continued (2)**
> >
> > ### **What do you mean by strictly lying on the diagonal line y=x? Wouldn’t that suggest that in-distribution and out-of-distribution agreement are of the same magnitude?**
> > We apologize for the confusion. In our CLIP linear probing experiments, we saw that with data subsetting and ordering, while ID versus OOD agreement was strongly linearly correlated, the linear fit did not match the fit of ID versus OOD accuracy, and lied strictly above it. Empirically, we see that the linear fit is instead very close to the diagonal y=x line. This is an observation, and by no means do we argue that this strict diagonal trend is always observable with data subsetting/ordering. We have de-emphasized this statement in our paper accordingly.
> >
> > &nbsp;
> >
> > ### **It remains a bit unclear to me to what degree this method needs to be first validated before the results can be trusted…Is the method significantly more stable to changes in the training protocol when training from scratch?**
> >
> > Thank you for the great question! These are all significant considerations we have attempted to address in our work.
> >
> > Prior work [1] shows that a set of neural networks, all trained from scratch, are able to exhibit the desired agreement-on-the-line (AGL). The phenomenon is much more stable in this regime. In some sense, models diverge significantly from initialization in this scenario. However, we see that in the case of fine-tuning foundation models, only one of those settings i.e. random initialization is able to produce AGL.
> >
> > Our finding that AGL is observable without training from scratch when the linear head is randomly initialized is quite robust. To reiterate, AGL refers to the phenomenon wherein the ID versus OOD agreements observe strong linear correlation, and the ID versus OOD accuracies are also strongly linearly correlated _with the same slope and bias_. In our paper, we observe this phenomena in more than 50 datasets and hundreds of models. With the random head initialized method studied in Section 3.1, AGL holds for CLIP linear-probe models over these datasets:
> > 1. CIFAR10 to CIFAR10C, CIFAR10.1
> > 2. CIFAR100 to CIFAR100C
> > 3. ImageNet to ImageNetC, ImageNetV2
> > 4. WILDS domain adaptation benchmark datasets, namely FMoW, iWildCam, and Camelyon17. When training models from scratch for iWildCam, agreement showed to be weakly correlated (see Section 3.3 of [1]), whereas with a linear probe over CLIP agreement shows strong linear correlation and matches the linear fit of accuracy.
> > The ACL-AGL plots for all of these datasets can be seen in Section 9.8.
> >
> > We have also expanded our same study to the **_newly added_**
> > 1. OfficeHome dataset, another domain adaptation benchmark consisting of 4 domains of images [Art, ClipArt, Product, Real] for 65 classes. The graphical results illustrating the accuracy-agreement trends can be found in Section 9.3, and we find yet again that random head is the only method consistently exhibiting AGL behavior.
> > 2. We also explore the domain of text classification, wherein we train linear-probes atop of GPT and OPT features, using the MNLI and SNLI datasets as ID and OOD respectively.
> >
> > Our finding, i.e. the requirement of randomly initialized linear heads, remains consistent across both modalities. Furthermore, this observation also extends to another training protocol, i.e. fully fine-tuning. As can be seen in Sections 3.2 and 9.4, the random heads are the best at producing AGL behavior for GPT fully fine-tuned models. Sections 9.5 results in the same findings for OPT and BERT full fine-tuned models. We also examine the **_newly added_** fully fine-tuned CLIP models in Section 9.3, and our results remain consistent.

---

> > > ### Author Response · Authors · 2023-11-17
> > > **Rebuttal Continued (3)**
> > >
> > > ## **Other questions**
> > > ### **It’s also not clear how significant a deviation of 5% is if the absolute values are not known e.g. if test performance is 20%, then a deviation of 5% is clearly more significant than if test performance is 95%.**
> > > Yes, this is a great point! In response to your feedback, we have added the mean absolute percentage error (MAPE) to our tables, instead of just reporting MAE, as the MAPE captures the relative not just absolute deviation from the true value.
> > > The following is a snapshot of the new metric for ALine and other metrics on the SQuAD-Shifts, CIFAR10-C and ImageNet-C datasets.
> > >
> > > |    Shift     |  ALine   | Naive Agreement |  ATC   |   AC   |   DF   |
> > > |--------------|----------|-----------------|--------|--------|--------|
> > > | SQuAD-Shifts | **1.68%**    | 19.48%          | 9.16%  | 45.04% | 4.54%  |
> > > | CIFAR10-C    | **6.92%**    | 44.33%          | 18.69% | 48.66% | 32.79% |
> > > | ImageNet-C   |  **10.91%** | 56.76%          | 27.24% | 79.00% | 37.86% |
> > >
> > > 1. We also show the MAE by OOD accuracy range in the Appendix (Section 9.12) for any readers interested in how the deviation breaks down by performance. We nevertheless see that the ALine estimation MAE only slightly increases with the true accuracy, and that this method doesn’t penalize low or high performing models.
> > > 2. In addition, we have added figures in the Appendix (Section 9.9) that present a comparison of OOD Accuracy vs Predicted Accuracy for ALineD and competing baselines to illustrate predictive performance at every test accuracy value. These plots further illustrate how ALineD is equally predictive of OOD performance at low test performance levels as it is at high performance levels. These are anonymized links to view these plots:
> > >     - SQuAD-Shifts Amazon [Fig 17a](https://i.imgur.com/XWnGUNE.png)
> > >     - SQuAD-Shifts Reddit [Fig 17b](https://i.imgur.com/3l8cati.png)
> > >     - SQuAD-Shifts New York Times - [Fig 17c](https://i.imgur.com/bEvPyYv.png)
> > >     - SQuAD-Shifts New Wiki [Fig 17d](https://i.imgur.com/pRgKExk.png)
> > >
> > >
> > > ### **What are the units of the x-axis and y-axis for Figure 1 and Figure 2? How is agreement measured? Is this a log-log scale? How are the linear fits obtained?**
> > > We are sorry for the confusion.
> > > 1. The units of the x and y axes in Figures 1 and 2 represent the common metric used to calculate accuracy and agreement, represented as a percentage (See Equation 1 and Section 2.1). To clarify this metric, for image and text classification, accuracy is calculated as the percent of correct predictions while for question answering, it is represented through the F1 score.
> > > 2. Agreement is computed between a pair of model predictions and, is an absolute number like accuracy. They differ in that while computing agreement, the second model’s prediction substitutes in place of the ground truth that is used while computing accuracy.
> > > 3. When plotting accuracy and agreement for a distribution shift, probit scaling ($\Phi^{-1}$) is applied to the accuracy and agreement values, making the axes probit scaled. This transformation has been shown to yield better linear fits [1,2]. Finally, linear fits are computed from these plots through closed form solutions of 1D linear regression. We provide details of this computation in Appendix Section 9.2.
> > >
> > >
> > > ### **Additional changes**
> > > 1. (Minor) We have removed hyperparameters from the four sources of diversity as the category is too generic for a controlled study. This is reflected in the paper as well. Thus, we have three sources of diversity: random head initialization, data shuffling, and data subsetting. We’d also like to emphasize that this does not change our results – random head initialization induces AGL most effectively.
> > > 2. We have changed the error metric of ALine in the main body from mean absolute error (MAE) to mean absolute percentage error (MAPE) as the MAPE captures the relative not just absolute deviation from the true value. However, we have also included the MAE by accuracy range in the Appendix (Section 9.12).
> > >
> > > ### **[References]**
> > > [1] Christina Baek, Yiding Jiang, Aditi Raghunathan, and J Zico Kolter. “Agreement-on-the-line: Predicting the performance of neural networks under distribution shift.” 2022.\
> > > [2] John P Miller, Rohan Taori, Aditi Raghunathan, Shiori Sagawa, Pang Wei Koh, Vaishaal Shankar, Percy Liang, Yair Carmon, and Ludwig Schmidt. “Accuracy on the line: on the strong correlation between out-of-distribution and in-distribution generalization.” 2021.\
> > > [3] Ananya Kumar, Aditi Raghunathan, Robbie Jones, Tengyu Ma, and Percy Liang. "Fine-tuning can distort pretrained features and underperform out-of-distribution." 2022.\
> > > [4] Preetum Nakkiran and Yamini Bansal. “Distributional generalization: A new kind of generalization.” 2020.

---

> > > > ### Author Response · Authors · 2023-11-21
> > > > **Any further questions?**
> > > >
> > > > Thank you again for the comprehensive and useful review! According to your suggestions, we've tried to clarify our usage of the term "diversity" throughout the paper and provided a more fine-grained comparison with baseline methods for estimating the performance models binned by their OOD accuracies (It's desirable to have smaller deviation for lower performing models). We hope this addresses your concerns.
> > > >
> > > > Do you have any further questions? If you have any more suggestions or feedback, please let us know.

---

### Official Review · Reviewer_47Ds · 2023-11-01

**Soundness:** 3 good
**Presentation:** 3 good
**Contribution:** 2 fair
**Rating:** 6
**Confidence:** 3

**Summary:**

This paper studies "agreement on the line" (AGL) phenomenon for the foundation model setting which can be used to predict out-of-distribution (OOD) accuracy without OOD labels. Roughly, AGL implies that for an appropriately chosen family of models, the in-distribution (ID) and OOD accuracies lie on a straight line (accuracy on line - ACL) and so does the ID and OOD agreements between pairs of models, and that these lines are the same. Since agreement for a family of models can be measured using ID and OOD unlabeled data, and ID accuracy can be measured using ID labeled data, one can predict OOD accuracy by estimating the AGL line.

This phenomenon has been previously studied and reported for the supervised learning setting. However the paper argues that AGL is challenging for foundation model setting:
- For light fine-tuning from a single foundation model, it is hard to get a diverse set of models to observe AGL
- For multiple foundation models, the set of models might be too diverse and AGL might fail

The paper puts AGL to test for (1) linear probe with CLIP on CIFAR & (2) fine-tuning with language model(s). It considers 4 different types of model families by varying (a) random head initialization, (b) hyperparameters, (c) data subsets, (d) data ordering. The main finding is that **"random head initialization" is the only setting that demonstrates AGL, for linear probe and finetuning.** Thus for this setting, careful selection of model family is necessary to observe AGL.

For the multiple foundation model setting, the paper finds that **AGL holds across a family of 41 language models** for QA finetuning, despite the LLMs being pretrained on different data sources. This contrasts earlier findings for pretrained models in vision setting. The presence of AGL allows better OOD accuracy prediction that other methods based on *confidence* based predictions.

**Strengths:**

- Originality of findings: It is an interesting finding that AGL also holds in fine-tuning setting, although with carefully selected model family. Also interesting to see that random head initialization is the best and only setting for which AGL holds. AGL across different pretrained LLMs is also interesting, given that the same doesn't hold for vision setting.

- Clarity: The high level message and results are clearly presented. Some details could be presented better; see comments/questions below.

**Weaknesses:**

- Novelty: The paper evaluates an existing idea of AGL using existing metrics, but in a different (but relevant) setting of foundation model fine-tuning and linear probing. It does not propose a new technique or a new evaluation method or a drastically new perspective, to the best of my knowledge. The findings are new mostly because of the new setting that is being considered.

- Clarity: Some details are either deferred to the appendix (definition of ALINE) or the reader is directed to prior work (methods that utilize model confidence in Section 4.2), which made it harder to follow some details. Figures captions could also use more details and be more self-contained (e.g.  Figure 3, Figure 4 -- which column is what method?). More comments/questions below

- While the paper presents findings on when AGL holds in the fine-tuning setting, there is not much insight into why these findings might hold. E.g. is there any intuition for why random head initialization leads to AGL and others do not?

Overall it seems like the results are "good to know", but do not necessarily provide a lot of new insights or food for thought. A deeper analysis of some of the findings could make the paper significantly stronger in my opinion. Given that I did not find any major flaws, I would assign a score of weak accept.

**Questions:**

- In Figure 4, which column corresponds to which method? Visually it seems like the second column is the best, and for no column does it seem like AGL always holds.

- In Section 4.1 the phrase “ensemble of foundation models” is a bit confusing. Does it mean an ensemble in the machine learning sense or just a "group" of models?

- In Section 4.2, how is temperature scaling relevant in this setting?

- What is $\Phi$ in Section 8.3?

- Is there a specific reason to only do linear probing for the CLIP setting, and just 1 fine-tuning task of SQuAD for the LM setting?

---

> ### Author Response · Authors · 2023-11-17
> **Thank you for your time and constructive feedback**
>
> Dear Reviewer,
>
> Thank you for your time and feedback! We have now made the following changes in response to your feedback:
> 1. We have added a paragraph in the introduction summarizing our main contributions and what insights we add compared to prior works
> 2. We have created a new Section 5 discussing the comparison between the ALine method and the other baseline methods.
> 3. We have made the table and image captions more descriptive.
> 4. We have expanded our study of CLIP for Image Classification to full fine-tuning, and shown that the random linear head initialization is still the only method consistently yielding ensembles exhibiting Agreement-on-the-line (see [Table 7](https://i.imgur.com/RmhQc3O.png)).
> 5. We have provided some preliminary intuition as to the specific advantage of random linear head initialization for obtaining ensembles which exhibit AGL in the response below. We could add it to the paper if it’s helpful.
>
> We address your concerns below in more detail:
>
> &nbsp;
>
> ### **Novelty and new perspectives**
> Thank you for your feedback! We agree that the main “novelty” in our study is that we study a new setting of fine-tuning foundation models. Methodologically, we do borrow heavily from prior work of Agreement-on-the-line [1], but we make some important changes to make it work in this setting. Prior work uses an ensemble of independently trained models, but this is intractable in the new setting. Therefore, we study three different ensembles that are tractable, and find that random head initialization is the only ensemble that works, but it works surprisingly well! We think our observations add the following new insights into the broader phenomenon of “agreement-on-the-line”, some of which surprisingly contradict hypotheses or suggestions made in prior works.
> 1. No prior work shows any distinction between the agreements afforded by different ways of generating ensembles—for example data subsetting, or different random initializations etc. We find that random head initialization is therefore a special method of introducing diversity. We hope our work inspires future theoretical studies on the effect of random head initialization.
> 2. [1] found that linear models when trained from scratch do not show AGL. However, we see that linear models on top of CLIP features show AGL, suggesting that it’s less a property of the model family and more a property of the data distribution. This observation also makes it more tractable to study AGL theoretically since linear models are easier to analyze than neural nets
> 3. Prior work in ACL [2, 5] suggested that different pre-training data for vision models would lead to different effective robustness i.e. the trend between ID and OOD performance. We find the contrary in Section 4, that different pretrained language models can observe the same robustness.
>
> &nbsp;
>
> ### **Intuition for why random head initialization is different**
> Our (admittedly imprecise) intuition is that since we are doing only a few steps of fine-tuning, we never move too far from the random head initialization. Hence with different initializations, we end up with models that can be quite “far” from each other, despite starting from the same foundation model. This paper [4] shows that random initialization dictates how the pretrained features end up changing, and provides some insight into why the head initialization can be so important. On the other hand, if we fix the random initialization, we might not end up moving too far, and as a result even with different data subsetting or data ordering, we might end up with very similar models that agree a lot (agreement line is much higher than accuracy line). If this explanation is helpful, we’d be happy to add it to our paper or discuss more.
>
> &nbsp;
>
> ### **Some details are either deferred to the appendix (definition of ALINE) or the reader is directed to prior work (methods that utilize model confidence in Section 4.2), which makes it harder to follow some details.**
> Thank you for the feedback! We have now added a sentence to summarize what ALine is doing in our baseline comparison section (Section 5).

---

> ### Author Response · Authors · 2023-11-17
> **Rebuttal Continued (2)**
>
> ### **Figures captions could also use more details and be more self-contained (e.g. Figure 3, Figure 4 -- which column is what method?).**
> Thanks for pointing this out, and apologies for the unwanted confusion. We have rewritten all of our table and image captions to briefly recap the experiment, the dataset and methodology, and the drawn conclusion. Specifically,
> 1. In Figure 3, we have updated our caption to -  _ACL and AGL observed on extractive Question Answering when computed over a set of models fine-tuned from different base foundation models. ACL and AGL are seen to hold for all four shifts of SQuAD-Shifts under this setup. The base models used here are OPT, GPT, and LLama_ .
> 2. Figure 4 has been moved to Figure 6 in the updated paper with the caption - _ID vs OOD trends of accuracy and agreement of LLMs fine-tuned for Question Answering from a single pretrained base model (GPT2-Medium). Each column presents trends for different sources of stochasticity employed to obtain a diverse ensemble of fine-tuned models. We see that random linear head initialization is the best method to obtain a set of models exhibiting AGL behavior._
>
> &nbsp;
>
> ## **Questions:**
>
> ### **In Figure 4, which column corresponds to which method? Visually it seems like the second column is the best, and for no column does it seem like AGL always holds.**
> We apologize for the confusion. We have added labels for Figure 4 (the new Figure 6), it is random head initialization, data shuffling, and data subsetting from left to right. AGL holds the best with random head initialization, and the respective ALine (a method which leverages this proximity of the lines for estimating accuracy) MAPE errors can be found in table 2, which reaffirms this.
>
> ### **In Section 4.1 the phrase “ensemble of foundation models” is a bit confusing. Does it mean an ensemble in the machine learning sense or just a "group" of models?**
> We apologize again for the confusion, we use ensemble here simply to refer to a collection of models. This usage is consistent with [1] and [3].
>
> ### **In Section 4.2, how is temperature scaling relevant in this setting?**
> We temperature scale all our baselines which are confidence based, and report the numbers with the best performance between with and without temperature scaling.
>
>
> ### **What is $\Phi$ in Section 8.3?**
> We apologize for the misclarification. $\Phi^{-1}$ refers to probit scaling which is applied to induce a better linear fit as in [1] and [2]. We have added the explanation in the Appendix (Section 9.2).
>
> ### **Is there a specific reason to only do linear probing for the CLIP setting, and just 1 fine-tuning task of SQuAD for the LM setting?**
>  Thanks for your question. We initially picked linear probing for CLIP since we wanted to consider the “lightest” form of fine-tuning. However, we have now added full fine-tuning as well.
> 1. See our full fine-tuning experiments for CLIP based Image classification on the CIFAR10 dataset in Appendix Section 9.3.
> 2. For SQuAD in the LM setting, linear probing resulted in poorly performing models and was hence omitted.
> 3. We have included Text classification in Appendix Section 9.6/9.7 where we perform both full-finetuning and linear probing. Overall, our finding of the need for random head initialization for AGL, holds across all modalities, datasets, and fine-tuning methods.
>
> We present a summary of our experiments below:
>
> **Image Classification with CLIP**
>
> * **Full Fine-tuning:**
>     1. (**_Newly added_**)  CIFAR10 Image classification: ID dataset: CIFAR10; OOD datasets: CIFAR10-C
> * **Linear Probing:**
>     1. CIFAR10 Image classification: ID dataset: CIFAR10; OOD datasets: CIFAR10-C, CIFAR10.1
>     2. (**_Newly added_**) OfficeHome: We use the standard domain adaptation benchmark with 4 domains: “Art”, “ClipArt”, “Product”, “Real”. We test all 3x3 settings where each domain is considered ID one at a time, with the remaining 3 domains being OOD
>     3. ImageNet: ID dataset: ImageNet, OOD datasets: ImageNetV2, ImageNetC
>     4. Satellite imagery: fMoW-WILDS
>     5. Species classification from camera traps: iWildCam-WILDS
>     6. Pathology: ID dataset: Camelyon17-WILDS
>
> ### **NLP tasks**
>
> * **Full Fine-tuning:**
>     1. Question answering: ID dataset: SQuAD; OOD datasets: (i) SQuAD-Shifts - Reddit (ii) SQuAD-Shifts - Amazon (iii) SQuAD-Shifts - NYT (iv) SQuAD-Shifts - New Wiki
>         - Single Base Model: GPT2-Medium, OPT-125M (**_Newly added_**), BERT (**_Newly added_**)
>         - Multiple Base Models: GPT2 variant, GPT-Neo, OPT variants and Llama2
>     2. (**_Newly added_**) Text classification: ID dataset: MNLI-matched, OOD datasets: (i) MNLI-mismatched (ii) SNLI (iii) WNLI (iv) HANS
> * **Linear Probing:**
>     1. (**_Newly added_**) Text classification: ID dataset: MNLI-matched, OOD datasets: SNLI

---

> ### Author Response · Authors · 2023-11-17
> **Rebuttal Continued (3)**
>
> ### **Additional changes**
> 1. (Minor) We have removed hyperparameters from the four sources of diversity as the category is too generic for a controlled study. This is reflected in the paper as well. Thus, we have three sources of diversity: random head initialization, data shuffling, and data subsetting. We’d also like to emphasize that this does not change our results – random head initialization induces AGL most effectively.
> 2. We have changed the error metric of ALine in the main body from mean absolute error (MAE) to mean absolute percentage error (MAPE) as the MAPE captures the relative not just absolute deviation from the true value. However, we have also included the MAE by accuracy range in the Appendix (Section 9.12).
>
> ### **[References]**
> [1] Christina Baek, Yiding Jiang, Aditi Raghunathan, and J Zico Kolter. “Agreement-on-the-line: Predicting the performance of neural networks under distribution shift.” 2022.\
> [2] John P Miller, Rohan Taori, Aditi Raghunathan, Shiori Sagawa, Pang Wei Koh, Vaishaal Shankar, Percy Liang, Yair Carmon, and Ludwig Schmidt. “Accuracy on the line: on the strong correlation between out-of-distribution and in-distribution generalization.” 2021.\
> [3] Yiding Jiang, Vaishnavh Nagarajan, Christina Baek, and J Zico Kolter. “Assessing generalization of sgd via disagreement. International Conference on Learning Representations” 2022.\
> [4] Ananya Kumar, Aditi Raghunathan, Robbie Jones, Tengyu Ma, and Percy Liang. "Fine-tuning can distort pretrained features and underperform out-of-distribution." 2022.\
> [5] John Miller, Karl Krauth, Benjamin Recht, and Ludwig Schmidt. “The effect of natural distribution shift on question answering models.” 2020.

---

> > ### Author Response · Authors · 2023-11-21
> > **Any further questions?**
> >
> > Thank you again for the comprehensive and useful review! According to your suggestions, we've made sections more self-contained and clarified our contributions (_our work, in addition to providing state-of-the-art performance estimation of large foundation models, debunks several hypotheses about agreement-on-the-line and accuracy-on-the-line from previous studies_). We've also provided some intuition for why random linear heads may be important to observe agreement-on-the-line, although we leave a more rigorous theoretical analysis for future work. We hope this addresses your concerns.
> >
> > Do you have any further questions? If you have any more suggestions or feedback, please let us know.

---

### Official Review · Reviewer_BrfQ · 2023-11-02

**Soundness:** 3 good
**Presentation:** 3 good
**Contribution:** 3 good
**Rating:** 6
**Confidence:** 4

**Summary:**

This paper focuses on the problem of predicting the performance of fine-tuned foundation models under distribution shifts. To do so, they extend ALine (Baek et al. 2022), a method that leverages accuracy-on-the-line and agreement-on-the-line phenomena. The main idea of this extension is to inject diversity in model performance of fine-tuned foundation models via random linear head initialization, or use multiple foundation models trained with different hyperparameters and on different datasets. The empirical findings demonstrate that this approach outperforms existing methods in terms of MAE (mean absolute error).

Update: the rebuttal addresses my main concerns, so I am increasing my score.

**Strengths:**

- Shows that multiple foundation models in vision and LLM-based tasks exhibit accuracy-in-the-line and agreement-on-the-line phenomena.
- Analysis on sources of diversity and its effect on ACL and AGL is thorough. This analysis results in a simple fix (random linear heads) to estimate OOD performance of a single foundation model fine-tuned on a downstream task.
- Experiments show that ALine-D and Aline-D (Baek et al., 2022) outperform OOD prediction methods based on model confidence in predicting OOD performance of fine-tuned foundation models.

**Weaknesses:**

- Organization of the paper is quite confusing. It starts off with a single model regime, where ALine needs to be extended due to diversity issues. Then, it states (in text) that multiple foundation models may have too much diversity, but empirically this is not a problem and that ALine works directly. It may be better to start with S4 (show that multiple foundation models exhibit AGL) and then move to the single foundation model setup that requires modifications to AGL-based OOD error estimation.
- The novelty of this work is limited. S5 applies ALine (Baek et al. 2022) to larger pre-trained (foundation) models. S4 extends ALine by training a single foundation models with multiple random linear head initializations.
- A major limitation of {accuracy, agreement}-on-the-line phenomena is that it is primarily a dataset-level property. First, there exist multiple datasets wherein the  ID-OOD ACL trend is not well explained by a linear function (see https://arxiv.org/abs/2209.00613 and https://arxiv.org/abs/2305.02995). Second, it is unclear when the ACL and AGL trend (i.e., same slope and intercept) hold. As a result, one cannot reliably use this method to predict OOD error in practice.
- Comparison to the ProjNorm method (https://arxiv.org/abs/2202.05834), which performs better than the baselines considered in this paper, is missing.

**Questions:**

It is unclear to me why injecting diversity via random linear heads works but not via other methods, e.g., data subsetting. Any intuition for what separates random linear heads from other interventions?

---

> ### Author Response · Authors · 2023-11-17
> **Thank you for your time and constructive feedback**
>
> Dear Reviewer
>
> Thank you very much for your time and helpful feedback! We have now made the following changes:
> 1. We have clarified the role of different sections in the response below.
> 2. Added a new baseline method, i.e. ProjNorm [5] (Appendix, Section 9.10), for OOD performance estimation for the Question-Answering model ensembles.
> 3. We have provided some preliminary intuition as to the specific advantage of random linear head initialization for obtaining ensembles which exhibit AGL in the response below. We could add it to the paper if it’s helpful. .
>
> ### **Paper Organization (Putting Section 3 with more contributions after Section 4)**
>
> Thank you for your feedback! We apologize for any confusion about the ordering of our contributions.
> 1. We’d like to clarify our reasoning for placing Section 3 before Section 4. Our primary focus was studying Agreement-on-the-line (AGL) in the setting wherein we fine-tune models from a single base foundation model. There may be many situations where we do not have access to multiple foundation models.
> 2. Furthermore, since several foundation models are “black-box” in different ways (we don’t always know what they are trained on, for how long, what architecture etc), there’s always the risk of models being too different to have their fine-tuned counterparts lie on the same ID-OOD accuracy line. Thus Section 4 was just to show an interesting independent and complementary observation that in some scenarios, where we do have access to multiple foundation models, AGL seems to hold when using this ensemble of existing models.
> 3. It is true that Section 3 requires us to be smart when creating the ensemble for evaluating AGL, while Section 4 allows us to use existing models directly. But we think Section 3 is the more general setting that’s of primary interest. We can make this more clear in our discussion of different sections and layout of the paper. Does this address your concern or do you still think we should reorder our sections?
>
>
> ### **Limited novelty**
> 1. We agree that in terms of **methodological advancements** our novelty might seem limited. However, we strongly believe that our paper makes **evaluative/conceptual** advancements to understand the phenomenon of agreement-on-the-line (AGL) in the setting of fine-tuning foundation models, where we do not have access to independently trained models from scratch.
> 2. As the reviewer notes, AGL is not a universal phenomenon, and we need to identify the right ensemble of models to use. Prior work takes a set of independently trained models from scratch, trained on the same dataset. This is not feasible in the setting of foundation models—we simply cannot train multiple foundation models on the same dataset. In fact, apriori we did not expect any form of AGL to hold with fine-tuned models from the same foundation model, because we expected their error to be highly correlated.
> 3. We perform a systematic evaluation of different ways to create ensembles of fine-tuned models, and find that only one method worked reliably across the board: having different initializations of the random head. We believe this is a finding of great practical relevance because it allows us to perform unsupervised accuracy estimation when using foundation models. Our method achieves state-of-the-art.
> 4. We also believe this work advances our understanding of the broader phenomenon of AGL. For example, [4] found that linear models when trained from scratch do not show AGL. However, we see that linear models on top of CLIP features show AGL, suggesting that it’s less a property of the model family and more a property of the data distribution.  Prior work in ACL [3] suggested that different pre-training data for vision models would lead to different effective robustness i.e. the trend between ID and OOD performance. We find the contrary in Section 4, that different pretrained language models can observe the same robustness.
>
> We hope that this discussion also clarifies the novelty and scientific value of our work. To summarize, we provide a rigorous study of limitations of applying AGL for foundation models. Methodologically, this also lends us SoTA estimation of the performance of foundation models. We have added a contributions section in the introduction to clarify the novelty and can further improve upon our writing for the camera-ready version. Thank you again for your time, please let us know if there is anything else you would suggest to help clarify our work.

---

> > ### Author Response · Authors · 2023-11-17
> > **Rebuttal Continued (2)**
> >
> > ### **A major limitation of {accuracy, agreement}-on-the-line phenomena is that it is primarily a dataset-level property. …First, there exist multiple datasets wherein the ID-OOD ACL trend is not well explained by a linear function (see https://arxiv.org/abs/2209.00613 and https://arxiv.org/abs/2305.02995). Second, it is unclear when the ACL and AGL trend (i.e., same slope and intercept) hold.**
> > Thanks for raising an important point; we hope to clarify below and also clarify in our draft.
> > 1. We agree with the reviewer that accuracy-on-the-line (ACL) is a dataset level property, but in practice, whenever ACL doesn’t hold, agreements also do not show a linear fit, as also noted in prior work [4]. Therefore, we can at least identify such failure cases using unlabeled data, and decide to not use agreement-on-the-line (AGL) for accuracy estimation. This should catch the failure cases the reviewer mentions.
> > 2. There’s the next concern that even if ACL holds, ACL and AGL need not always have the same slope and intercept. This is one of the findings in our paper, it is non-trivial to make AGL hold when lightly fine-tuning a single foundation model. For example, ensembles created with different data subsets or different data orderings exhibit ACL but agreement slopes and biases do not match that of accuracy, i.e. AGL does not hold. We find that random head initialization consistently yields ensembles exhibiting AGL, across all the settings we tried. But to reiterate; we do not claim that random head initializations would always lead to AGL (i.e. the matching slope and bias between agreement and accuracy). But this is not unexpected, as there is provably no single optimal performance estimation method [1]. Any estimation method has to rely on certain assumptions about the distribution shift. The purpose of our work is not to argue for the universal application of AGL or ACL, but to show that it can be a useful tool across a variety of settings when used with the right ensembles. It is interesting future work to characterize theoretically when exactly AGL (or ACL) holds.
> >
> > ### **Comparison to the ProjNorm method (https://arxiv.org/abs/2202.05834), which performs better than the baselines considered in this paper, is missing.**
> > Thank you for the feedback! As per suggestion, we have added the ProjNorm baseline to the paper in Section 9.10 of the Appendix.
> > 1. ProjNorm yields a score that is shown to be well correlated with OOD accuracy. Unlike ALine and other baselines, this score is only amenable for ranking the OOD performance of a set of models rather than directly predicting their OOD accuracy values. As a result, comparisons with ProjNorm are presented in terms of the strength of the linear correlation between a method’s output and the true OOD accuracy.
> > 2. We find that ProjNorm on its own yields very poor correlation (R2 < 0.01). On further investigation, we find that ProjNorm of certain model families (OPT-125m) is completely uncorrelated with their OOD accuracy. After removing such models, ProjNorm shows an average R2 of 0.70 with the OOD accuracies of models fine-tuned from different bases, which is still lower than using ALineD (average R2 = 0.98) in this setting.
> > 3. In the paper, we present ProjNorm values after removing uncorrelated models. At present we have computed ProjNorm on 15 out of the 41 models evaluated in Section 4. We will make sure to evaluate all models for the final version of this paper, but we do not expect our findings to change significantly.
> >
> > ### **Any intuition for what separates random linear heads from other interventions?**
> > This is indeed a surprising observation, and we think it would be interesting future work to understand this precisely. In particular, we find this observation to be true even for linear probing on CLIP features, which is potentially amenable to theoretical analysis. Our (admittedly imprecise) intuition is that since we are doing only a few steps of fine-tuning, we never move too far from the random head initialization. Hence with different initializations, we end up with models that can be quite “far” from each other, despite starting from the same foundation model. This paper [2] shows that random initialization dictates how the pretrained features end up changing, and provides some insight into why the head initialization can be so important. On the other hand, if we fix the random initialization, we might not end up moving too far, and as a result even with different data subsetting or data ordering, we might end up with very similar models that agree a lot (i.e. the agreement line lies much higher than the accuracy line). If this is helpful to add, we can add it to our paper.

---

> > > ### Author Response · Authors · 2023-11-17
> > > **Rebuttal Continued (3)**
> > >
> > > ### **Additional changes**
> > > 1. (Minor) We have removed hyperparameters from the four sources of diversity as the category is too generic for a controlled study. This is reflected in the paper as well. Thus, we have three sources of diversity: random head initialization, data shuffling, and data subsetting. We’d also like to emphasize that this does not change our results – random head initialization induces AGL most effectively.
> > > 2. We have changed the error metric of ALine in the main body from mean absolute error (MAE) to mean absolute percentage error (MAPE) as the MAPE captures the relative not just the absolute deviation from the true value. We have also included the MAE by accuracy range in the Appendix (Section 9.12).
> > >
> > > ### **[References]**
> > > [1] Saurabh Garg, Sivaraman Balakrishnan, Zachary C Lipton, Behnam Neyshabur, and Hanie Sedghi. “Leveraging unlabeled data to predict out-of-distribution performance.” 2022.\
> > > [2] Ananya Kumar, Aditi Raghunathan, Robbie Jones, Tengyu Ma, and Percy Liang. "Fine-tuning can distort pretrained features and underperform out-of-distribution." 2022.\
> > > [3] John Miller, Karl Krauth, Benjamin Recht, and Ludwig Schmidt. “The effect of natural distribution shift on question answering models.” 2020.\
> > > [4] Christina Baek, Yiding Jiang, Aditi Raghunathan, and J Zico Kolter. “Agreement-on-the-line: Predicting the performance of neural networks under distribution shift.” 2022.\
> > > [5]  Yaodong Yu, Zitong Yang, Alexander Wei, Yi Ma, and Jacob Steinhardt. “Predicting out-of-distribution error with the projection norm.” 2022

---

> ### Author Response · Authors · 2023-11-21
> **Any further questions?**
>
> Thank you again for the comprehensive and useful review! As you've suggested, we had incorporated more baselines (i.e. ProjNorm) and clarified our contributions (_our work, in addition to providing state-of-the-art performance estimation of large foundation models, debunks several hypotheses about agreement-on-the-line and accuracy-on-the-line from previous studies_). We've also provided some intuition for why random linear heads may be important to observe agreement-on-the-line, although we leave a more rigorous theoretical analysis for future work. We hope this addresses your concerns.
>
> Do you have any further questions? If you have any more suggestions or feedback, please let us know.

---

### Official Review · Reviewer_vvH6 · 2023-11-08

**Soundness:** 2 fair
**Presentation:** 3 good
**Contribution:** 3 good
**Rating:** 6
**Confidence:** 3

**Summary:**

This paper explores various methodologies such as commonsense reasoning, medical image classification adaptation, and robust fine-tuning of zero-shot models. The paper contributes to the field by presenting a real-world dataset for medical image classification, discussing the challenges of model adaptation to out-of-distribution data, and evaluating the performance of fine-tuned models on new, unseen datasets. It also examines the correlation between in-distribution and out-of-distribution generalization, offering insights into the predictability of model performance across different domains. This work stands to impact the understanding of model robustness and the practical application of transfer learning in diverse machine learning tasks.

**Strengths:**

The paper's approach to applying 'agreement-on-the-line' to foundation models is an original contribution that extends the utility of these models in novel ways. This method, as the paper suggests, could offer a fresh perspective on assessing and improving the performance of foundation models on out-of-distribution data. The authors' choice to explore this within the context of medical image classification and other domains also demonstrates an innovative application of machine learning techniques to real-world problems.

The authors' development and use of a real-world dataset for medical image classification suggest a commitment to grounding their findings in practical, applicable scenarios. The paper's methodological rigor is evident in the detailed descriptions of the experiments and the analytical techniques employed to assess model performance.


Despite the complex nature of the subject matter, the paper maintains a level of clarity that is commendable. The authors have managed to explain their methodologies and findings in a way that is understandable, which is particularly important when addressing such advanced topics in AI. The clarity with which the paper discusses the implications of its findings for the field of machine learning is a notable strength.

The significance of the paper's contributions cannot be overstated. By addressing the challenge of model generalization and robustness, the paper tackles one of the most pressing issues in machine learning today. The potential impact of this research is broad, as it could influence a wide range of applications, from healthcare to autonomous systems, where the ability to perform well on out-of-distribution data is crucial.

In summary, the paper stands out for its original approach to a key problem in machine learning, its rigorous and quality research methodology, the clarity of its exposition, and the potential significance of its contributions to the field.

**Weaknesses:**

The paper's exploration of foundation models and their application to various domains is commendable; however, there are areas where the work could be strengthened:

Specificity of Contributions:
The paper's contributions could be articulated more clearly. While the authors propose the application of the 'agreement-on-the-line' method to foundation models, they do not sufficiently differentiate this approach from existing methods. For instance, the paper states, "We consider a variety of foundation models: GPT2, GPT-Neo, OPT, Llama2, and CLIP," but does not elaborate on how 'agreement-on-the-line' enhances or differs from the current state-of-the-art. To improve, the authors should explicitly state the unique advantages and contributions of their method over existing approaches, possibly by providing a direct comparison to highlight the novelty.

Comparative Analysis:
The experimental section lacks a comprehensive comparative analysis. The authors present experiments validating their method, yet there is no benchmarking against existing Out-Of-Distribution (OOD) performance estimation methods. The paper could be significantly improved by including comparisons with established baselines, as this would demonstrate the efficacy of the 'agreement-on-the-line' method over others. For example, when discussing the fine-tuning procedures, the authors could compare the OOD performance estimation with other known approaches to establish the superiority of their method.

Data Diversity and Volume:
The volume and variety of datasets used for validation appear limited. The paper mentions, "Fine-tuning... we have access to labeled data from some distribution DID," but does not provide extensive validation across a broad range of datasets. Expanding experiments to include a wider array of datasets, especially those with larger scales and varying types, would lend more credibility and generalizability to the findings.

Writing Quality:
The clarity and organization of the paper could be improved. The logical flow and language precision are areas where the paper seems to fall short. For instance, the use of terms like "foundation models" and "agreement-on-the-line" could be more clearly defined to avoid ambiguity. The authors are encouraged to refine the language and structure of the paper to enhance readability and ensure that the arguments are presented coherently.

Reference Breadth:
The paper seems to have a narrow scope of references, primarily citing a few articles by the authors themselves. To establish the research within the broader context of the field, it would be beneficial to cite a wider range of high-quality, related studies. This would not only position the paper within the existing body of knowledge but also provide a more robust background for readers.

In summary, while the paper presents interesting ideas, it would benefit from clearer articulation of its unique contributions, more extensive comparative analysis, broader and more diverse data validation, improved writing quality, and a more comprehensive set of references.

**Questions:**

Methodological Clarification:
Could you elaborate on the theoretical underpinnings of the 'agreement-on-the-line' method? How does it theoretically and practically differ from existing methods for assessing model performance on out-of-distribution data?

Experimental Comparisons:
The paper would benefit from a direct comparison of your method with existing benchmarks. Could you include such a comparison to highlight the advantages of your approach?

Dataset Diversity and Volume:
Your experiments seem to be limited to a few datasets. Could you provide insights into how your method performs across a more diverse range of datasets, including those with larger scales?

Robustness of Findings:
How robust are your findings to changes in the model architecture or dataset characteristics? Are there any limitations to the applicability of the 'agreement-on-the-line' method?

Impact of Fine-Tuning Procedures:
Can you discuss the impact of different fine-tuning procedures on the performance of foundation models using your method? How does the 'agreement-on-the-line' adapt to these variations?

**Details Of Ethics Concerns:**

How were the datasets, especially those involving medical images, obtained? Were appropriate consents from the subjects obtained, and were the datasets de-identified to protect privacy?

---

> ### Author Response · Authors · 2023-11-17
> **Thank you for your time and constructive feedback**
>
> Dear reviewer,
>
> Thank you very much for your time and thorough feedback! We have now made the following changes which we believe address all your concerns and strengthen the paper
> - Added a paragraph in the introduction clearly describing our contributions and relation to prior work and state of the art
> - Clearly mention where we compare to baselines ([Table 3](https://i.imgur.com/rlCRcjC.png)), and added one more baseline method of ProjNorm [4]
> - Added results on OfficeHome dataset (with 4ID and 3 OOD settings for each) for image classification with CLIP, and MNLI to MNLI-Mismatched, SNLI, HANS, WNLI for text classification (See Appendix Sections [9.6](https://i.imgur.com/7RaLfqO.png), [9.7](https://i.imgur.com/HcF41ED.png)) . This now makes a total of 34 different distribution shifts for image classification and 8 different shifts in NLP tasks.
> - Considered two new architectures (OPT, BERT) in addition to GPT-2  and Llama for NLP tasks  (see [Table 8 and 9](https://i.imgur.com/qMePdYh.png))
> - Studied different kinds of fine-tuning: newly added full fine-tuning of CLIP ([Figure 5](https://i.imgur.com/TSy1Gdn.png)) in addition to linear probing
>
> We elaborate on each of your concerns below:
>
> ### **Specificity of Contributions: …'agreement-on-the-line' enhances or differs from the current state-of-the-art. To improve, the authors should explicitly state the unique advantages and contributions of their method over existing approaches, possibly by providing a direct comparison to highlight the novelty.**
>
> Thank you, to clarify our contributions, we modified the Introduction and added a “contribution” paragraph at the end. To summarize:
> - We propose a new state-of-the-art method for unsupervised accuracy estimation under distribution shifts when using large pre-trained models (a.k.a. foundation models) that are lightly fine-tuned for specific tasks. Prior works have primarily dealt with the models trained from scratch, where there is no transfer learning or extensive pre-training to account for. Hence, we believe the setting studied is new, and extremely relevant in today’s context. Prior work does not directly apply in this setting.
> - Furthermore, our work leveraging Agreement-on-the-line (AGL) for OOD estimation, builds on top of prior work [11]; but extends it in important ways to make it work in this new and important setting. Prior works have primarily dealt with the models trained from scratch, where there is no transfer learning or extensive pre-training to account for. Hence, we believe the setting studied is new, and extremely relevant in today’s context. Admittedly, it is similar in overall methodology—we compute the agreement between pairs in an ensemble of models. However, the key to making AGL work is obtaining the _right ensemble_. In [11], this ensemble was just a collection of models trained _independently from scratch_. A likewise (but infeasible) extension to pre-trained models would require numerous such models, all trained from scratch. Our work shows how to side-step this, by systematically identifying a computationally tractable method of obtaining the right ensemble. Specifically, we show that creating an ensemble with randomly initialized final linear heads and then fine-tuning can allow for AGL behavior, and apply downstream methods for unsupervised accuracy estimation, while other similar forms of ensembling (such as data ordering or data subsetting) do not.
> - We believe this is a significant finding of practical relevance, but also points to several interesting phenomena underlying AGL that go beyond previous knowledge. Prior work (section 3.4 in [11]) claimed that AGL does not hold for linear models. However, we find the contrary when using pre-trained features (such as CLIP). Furthermore, other prior work [14] suggests that the effective robustness (i.e. the linear fit b/w ID and OOD accuracy) would change depending on the pretraining data. We find that this is not the case for question answering with different pretrained LLMs. Thus we hope our findings can also advance our understanding of the robustness of ML models, particularly those that leverage foundation models.

---

> ### Author Response · Authors · 2023-11-17
> **Rebuttal Continued (2)**
>
> ### **Comparative Analysis: …no benchmarking against existing Out-Of-Distribution (OOD) performance estimation methods…**
>
> We apologize for any confusion about the lack of baselines. Our submission does compare to other baselines (Naive Agreement, ATC, AC, DF) in Table 3. We apologize for the table being in Section 4.2, which would have been confusing. The table reports numbers we have calculated over both models with randomly initialized heads (Section 3) and models from different bases (Section 4).
>
> In summary, ATC [1], AC [2], and DF [3] utilize model confidence to estimate OOD accuracy while Naive Agreement [9, 10]  directly utilizes agreement to predict accuracy. We have also added a new baseline of ProjNorm [4] in Section 9.10 which yields a score correlated with the OOD performance of the model. We see that our proposed method significantly outperforms baselines. The table below is a snapshot of some of our baseline comparisons; a full picture can be found in [Table 3](https://i.imgur.com/rlCRcjC.png). Based on your feedback, we have moved our baseline comparison section to its own standalone Section 5, and provided more details about the evaluated datasets, models, and baselines. Please let us know if there are any other baselines we should add.
>
> |    Shift     |  ALine   | Naive Agreement |  ATC   |   AC   |   DF   |
> |--------------|----------|-----------------|--------|--------|--------|
> | SQuAD-Shifts | **1.68%**    | 19.48%          | 9.16%  | 45.04% | 4.54%  |
> | CIFAR10-C    | **6.92%**    | 44.33%          | 18.69% | 48.66% | 32.79% |
> | ImageNet-C   |  **10.91%** | 56.76%          | 27.24% | 79.00% | 37.86% |
> &nbsp;
>
> ### **Data Diversity … The volume and variety of datasets used for validation appear limited**
>
> We list below the different datasets we evaluate on, including some that we added based upon your feedback (let us know if you feel this helps to address your concerns, and if there are particular new datasets that would be valuable to add in your opinion).
>
> **Image Classification with CLIP**
> 1. CIFAR10 Image Classification: ID dataset: CIFAR10; OOD datasets: CIFAR10-C (which includes 19 different shifts; we test on each individually), CIFAR10.1
> 2. (**_Newly added_**) OfficeHome: We use the standard domain adaptation benchmark with 4 domains: “Art”, “ClipArt”, “Product”, “Real”. We test all 4x3 settings where each domain is considered ID one at a time, with the remaining 3 domains being OOD
> 3. ImageNet: ID dataset: ImageNet, OOD datasets: ImageNetV2, ImageNetC
> 4. Satellite imagery: ID/OOD dataset: subsets of fMoW (WILDS)
> 5. Species classification from camera traps: ID/OOD dataset: iWildCam (WILDS)
> 6. Pathology: ID/OOD dataset: Camelyon-17 (WILDS)
>
> **NLP tasks with GPT, Bert, OPT, Llama**
> 1. Question answering: ID dataset is SQuAD; OOD datasets are (i) SQuAD-Shifts - Reddit (ii) SQuAD-Shifts - Amazon (iii) SQuAD-Shifts - NYT (iv) SQuAD-Shifts - New Wiki
> 2. (**_Newly added_**) Text classification: ID dataset is MNLI-matched, OOD datasets are (i) MNLI-mismatched (ii) SNLI (iii) WNLI (iv) HANS
>
> Our findings on when AGL occurs holds quite robustly across all these settings; i.e. random head initialization while fine-tuning is the only method consistently yielding ensembles where AGL holds, thus allowing for accurate OOD estimation with ALine methods.

---

> > ### Author Response · Authors · 2023-11-17
> > **Rebuttal Continued (3)**
> >
> > ### **Robustness [of AGL] Are there any limitations to the applicability of the 'agreement-on-the-line' method? How robust are your findings to changes in the model architecture or dataset characteristics?**
> >
> > Thank you for this important question!
> > 1. To clarify, “agreement-on-the-line” (AGL) itself does not hold universally across all distribution shifts, and even for the applicable ones, it doesn’t hold for every ensemble of models we constructed. **Our findings precisely study when it holds**. We identify that AGL does NOT hold when using ensembles that are fine-tuned from the same pretrained model (i.e. same base foundation model) when using different data subsets or data ordering without changing the head initialization. On the other hand, we see that by randomizing the head initialization, we are able to create an ensemble where AGL holds across multiple tasks (image-/text- classification, and question answering), multiple datasets (see Table 3), and multiple fine-tuning techniques (linear probing, full fine-tuning, and LoRA).
> > 2. When considering model architectures, we conducted **additional experiments with OPT-125M and BERT** for question answering in Appendix Section 9.5. Broadly, AGL holds for random head initialization. As seen from the table below, the MAPE of random head initialization is lower than when sets of models are obtained with data shuffling and subsetting, regardless of the model architecture.
> >
> > |          Setting           | GPT2-Medium | OPT-125M |  BERT  |
> > |----------------------------|-------------|----------|--------|
> > | Random Head Initialization | **3.48%**       | **5.43%**    | **8.62%**  |
> > | Data Shuffling             | 9.59%       | 8.70%    | 9.16%  |
> > | Data Subsets               | 13.94%      | 9.65%    | 18.64% |
> >
> >
> > 3. Our experiments span multiple modalities and datasets:
> > Image classification: CIFAR10, CIFAR100, ImageNet, WILDS (iWildCam, Camelyon17, fMoW)
> >     * (**_Newly added_**): OfficeHome
> >     * (**_Newly added_**) Text classification: MNLI
> >     * Question answering: SQuAD
> >
> > &nbsp;
> > ### **Impact of different finetuning procedures: Can you discuss the impact of different fine-tuning procedures on the performance of foundation models using your method?**
> >
> > Thank you for your feedback.
> > 1. As per your suggestion, we have expanded our results to now contain two different fine-tuning methods: both full fine-tuning and linear probing on CLIP.
> > 2. We also extend the study of these two methods to Text classification (Appendix Section 9.6) using different foundation models (GPT2-Medium and OPT-125M)
> > 3. Overall, we find that our main findings continue to hold across both procedures — ensembles generated via random head initializations show agreement on the line, and ensembles generated via other forms such as data subsetting or data ordering do not.
> >
> > For question answering, we consider models fine-tuned both via full fine-tuning and low rank adaptation. We see similar trends across both methods (Figure 3 includes models trained via both procedures). We do not test with linear probing for question answering because we found that linear probing achieved very low accuracies for this task.
> >
> > &nbsp;
> > ### **To establish the research within the broader context of the field, it would be beneficial to cite a wider range of high-quality, related studies.**
> > Thank you for your feedback!  We have a dedicated subsection 2.2 for scoping previous unsupervised performance estimation methods where we cover a broad range of methods from uniform-convergence based upper bounds, methods based on model confidence, and empirical estimates that probe the model’s performance on auxiliary tasks and data augmentations, In Subsection 2.3, we cite previous works on accuracy/agreement on the line. According to your suggestions, we made the following additions
> > 1. [8,9] also study the reliability of foundation models through designing OOD benchmarks that capture various failure modes.
> >
> > 2. Understanding how to gauge the reliability of foundation models has been of growing interest.
> > The relationship between agreement and _in-distribution_ accuracy have been established by [10, 12, 13]. Also related are Bayesian uncertainty estimation methods such as [5,6] that utilize the variance of model predictions to estimate model uncertainty.
> > 3. Recently, [7] provides some theoretical underpinning of AGL in the regression setting.

---

> > > ### Author Response · Authors · 2023-11-17
> > > **Rebuttal Continued (4)**
> > >
> > > ### **[References]**
> > > [1] Saurabh Garg, Sivaraman Balakrishnan, Zachary C Lipton, Behnam Neyshabur, and Hanie Sedghi. “Leveraging unlabeled data to predict out-of-distribution performance.” 2022.\
> > > [2] Dan Hendrycks and Kevin Gimpel. “A baseline for detecting misclassified and out-of-distribution examples in neural networks.” International Conference on Learning Representations, 2017.\
> > > [3] Devin Guillory, Vaishaal Shankar, Sayna Ebrahimi, Trevor Darrell, and Ludwig Schmidt. “Predicting with confidence on unseen distributions.” 2021.\
> > > [4] Yaodong Yu, Zitong Yang, Alexander Wei, Yi Ma, and Jacob Steinhardt. “Predicting out-of-distribution error with the projection norm.” 2022.\
> > > [5] Yarin Gal, Zoubin Ghahramani. “Dropout as a Bayesian Approximation: Representing Model Uncertainty in Deep Learning.” 2016. \
> > > [6] Balaji Lakshminarayanan, Alexander Pritzel, and Charles Blundell. "Simple and scalable predictive uncertainty estimation using deep ensembles." 2017.\
> > > [7] Donghwan Lee, Behrad Moniri, Xinmeng Huang, Edgar Dobriban, and Hamed Hassani. "Demystifying Disagreement-on-the-Line in High Dimensions." 2023.\
> > > [8] Andrey Malinin, Neil Band, German Chesnokov, Yarin Gal, Mark JF Gales, Alexey Noskov, Andrey Ploskonosov et al. "Shifts: A dataset of real distributional shift across multiple large-scale tasks." 2021.\
> > > [9] Dustin Tran, Jeremiah Liu, Michael W. Dusenberry, Du Phan, Mark Collier, Jie Ren, Kehang Han et al. "Plex: Towards reliability using pretrained large model extensions." 2022.\
> > > [10] Yiding Jiang, Vaishnavh Nagarajan, Christina Baek, and J Zico Kolter. “Assessing generalization of sgd via disagreement. International Conference on Learning Representations.” 2022.\
> > > [11] Christina Baek, Yiding Jiang, Aditi Raghunathan, and J Zico Kolter. “Agreement-on-the-line: Predicting the performance of neural networks under distribution shift.” 2022.\
> > > [12] Omid Madani, David Pennock, and Gary Flake. “Co-validation: Using model disagreement on unlabeled data to validate classification algorithms.” 2004. \
> > > [13] Preetum Nakkiran and Yamini Bansal. “Distributional generalization: A new kind of generalization.” 2021.\
> > > [14] John P Miller, Rohan Taori, Aditi Raghunathan, Shiori Sagawa, Pang Wei Koh, Vaishaal Shankar, Percy Liang, Yair Carmon, and Ludwig Schmidt. “Accuracy on the line: on the strong correlation between out-of-distribution and in-distribution generalization.” 2021.

---

> ### Author Response · Authors · 2023-11-21
> **Any further questions?**
>
> Thank you again for the comprehensive and useful review! As you've suggested, we had provided experiments on additional benchmarks (e.g. OfficeHome, MNLI), architectures (e.g. BERT, OPT, Llama), and reworded our contributions. We hope this addresses your concerns.
>
> Do you have any further questions? If you have any more suggestions or feedback, please let us know.

---

### Meta-Review · Area_Chair_tzmQ · 2023-12-02

**Metareview:**

This paper investigates the phenomenon of agreement on the line (AGL) for foundation model / LLMs finetuning. AGL is a phenomenon previously found in models trained from scratch, where for an appropriately chosen family of models, the in distribution and out of distribution classification accuracy is linearly correlated across models, i.e., the accuracies lie on a straight line, or accuracy on line (ACL). The ID and OOD agreement between pairs of models is also linearly correlated. With this phenomenon, OOD accuracy can be estimated by the AGL lines with ID accuracy, which is usually measurable with labeled data.
This work finds that with proper diversity introduced to model finetuning, the AGL/ACL phenomenon can also be observed. Specifically, the author tested 4 variations of diversity introduced by (1) random head initialization, (2) hyperparameters, (3) data subsets, (4) data ordering, and showed (1) demonstrates AGL for linear probe and finetuning CLIP on CIFAR dataset. The paper further shows AGL holds across a family of 41 language models for QA finetuning.

Strength:
 - The findings are novel and meaningful for ICLR community. Especially with the latest trend of foundation model and LLMs, fine-tuning on pretrained models dominates the learning paradigm compared to training from scratch. The finding of AGL on foundation model fine-tuning can open new research opportunities.
 - The AGL phenomenon seems working well if tuned properly (e.g., chose the right model family or the right way to introduce diversity).
 - AGL seems a general phenomenon across modalities (i.e., CLIP in vision language and LLMs in text are explored).

Weakness:
 - This work is an extension of previous work (Baek et al. 2022) to foundation models, which is somewhat limited its novelty in methodology. It's arguable that applying an existing methodology to a new domain (especially a trendy domain like LLMs) is still a good contribution. However, I feel this paper is too empirical (i.e., focusing on trying different model families or various ways to introduce model diversity) and lack of insight/explanation. Without support by explanation/theory/insight, the works make me wonder if the results are generalizable or just carefully tuned.

**Justification For Why Not Higher Score:**

I think this paper is at the border line. As I mentioned in the weakness, I do worry that the nice AGL phenomenon just comes from cherry pick and the lack of insight/explanation makes me wonder the generalizability of this work. However, the AGL phenomenon for foundation model finetuning is interesting if it does generalizable and I can see it's impact. Thus I am at the borderline.

**Justification For Why Not Lower Score:**

N/A

---

### Decision · Program_Chairs · 2024-01-16

Reject